# Operator Learning with Neural Fields: Tackling PDEs on General Geometries

**Louis Serrano[1], Lise Le Boudec[\*, 1], Armand Kassaï Koupaï[\*, 1], Thomas X Wang[1],**
**Yuan Yin[1], Jean-Noël Vittaut[2], Patrick Gallinari[1, 3]**
[1] Sorbonne Université, CNRS, ISIR, 75005 Paris, France
[2] Sorbonne Université, CNRS, LIP6, 75005 Paris, France
[3] Criteo AI Lab, Paris, France
{louis.serrano,lise.leboudec,armand.kassai,thomas.wang,yuan.yin
jean-noel.vittaut,patrick.gallinari}@sorbonne-universite.fr

## Abstract

Machine learning approaches for solving partial differential equations require learning mappings between function spaces. While convolutional or graph neural networks are constrained to discretized functions, neural operators present a promising milestone toward mapping functions directly. Despite impressive results they still face challenges with respect to the domain geometry and typically rely on some form of discretization. In order to alleviate such limitations, we present CORAL, a new method that leverages coordinate-based networks for solving PDEs on general geometries. CORAL is designed to remove constraints on the input mesh, making it applicable to any spatial sampling and geometry. Its ability extends to diverse problem domains, including PDE solving, spatio-temporal forecasting, and geometry-aware inference. CORAL demonstrates robust performance across multiple resolutions and performs well in both convex and non-convex domains, surpassing or performing on par with state-of-the-art models.

## 1   Introduction

Modeling physics dynamics entails learning mappings between function spaces, a crucial step in formulating and solving partial differential equations (PDEs). In the classical approach, PDEs are derived from first principles, and differential operators are utilized to map vector fields across the variables involved in the problem. To solve these equations, numerical methods like finite elements, finite volumes, or spectral techniques are employed, requiring the discretization of spatial and temporal components of the differential operators (Morton & Mayers, 2005; Olver, 2014).

Building on successes in computer vision and natural language processing (Krizhevsky et al., 2017; He et al., 2016; Dosovitskiy et al., 2021; Vaswani et al., 2017), deep learning models have recently gained attention in physical modeling. They have been applied to various scenarios, such as solving PDEs (Cai et al., 2021), forecasting spatio-temporal dynamics (de Bézenac et al., 2019), and addressing inverse problems (Allen et al., 2022). Initially, neural network architectures with spatial inductive biases like ConvNets (Long et al., 2018; Ibrahim et al., 2022) for regular grids or GNNs (Pfaff et al., 2021; Brandstetter et al., 2022b) for irregular meshes were explored. However, these models are limited to specific mesh points and face challenges in generalizing to new topologies. The recent trend of neural operators addresses these limitations by modeling mappings between functions, which can be seen as infinite-dimensional vectors. Popular models like DeepONet (Lu et al., 2022) and Fourier Neural Operators (FNO) (Li et al., 2022b) have been applied in various domains. However, they still have design rigidity, relying on fixed grids during training and inference, which limits their use in real-world applications involving irregular sampling grids or new geometries. A variant of

FNO tailored for more general geometries is presented in (Li et al., 2022a), but it focuses on design tasks.

To overcome these limitations, there is a need for flexible approaches that can handle diverse geometries, metric spaces, irregular sampling grids, and sparse measurements. We introduce CORAL, a COordinate-based model for opeRAtor Learning that addresses these challenges by leveraging implicit neural representations (INR). CORAL encodes functions into compact, low-dimensional latent spaces and infers mappings between function representations in the latent space. Unlike competing models that are often task-specific, CORAL is highly flexible and applicable to various problem domains. We showcase its versatility in PDE solving, spatio-temporal dynamics forecasting, and design problems.

Our contributions are summarized as follows:

- CORAL can learn mappings between functions sampled on an irregular mesh and maintains consistent performance when applied to new grids not seen during training. This characteristic makes it well-suited for solving problems in domains characterized by complex geometries or non-uniform grids.

- We highlight the versatility of CORAL by applying it to a range of representative physical modeling tasks, such as initial value problems (IVP), geometry-aware inference, dynamics modeling, and forecasting. Through extensive experiments on diverse datasets, we consistently demonstrate its state-of-the-art performance across various geometries, including convex and non-convex domains, as well as planar and spherical surfaces. This distinguishes CORAL from alternative models that are often confined to specific tasks.

- CORAL is fast. Functions are represented using a compact latent code in CORAL, capturing the essential information necessary for different inference tasks in a condensed format. This enables fast inference within the compact representation space, whereas alternative methods often operate directly within a higher-dimensional representation of the function space.

## 2 Related Work

**Mesh-based networks for physics.** The initial attempts to learn physical dynamics primarily centered around convolutional neural networks (CNNs) and graph neural networks (GNNs). Both leverage discrete convolutions to extract relevant information from a given node's neighborhood Hamilton (2020). CNNs expect inputs and outputs to be on regular grid. Their adaptation to irregular data through interpolation (Chae et al., 2021) is limited to simple meshes. GNNs work on irregular meshes (Hamilton et al., 2017; Veličković et al., 2018; Pfaff et al., 2021) and have been used e.g. for dynamics modeling (Brandstetter et al., 2022b) or design optimization (Allen et al., 2022). They typically select nearest neighbors within a small radius, which can introduce biases towards the type of meshes seen during training. In Section 4.2, we show that this bias can hinder their ability to generalize to meshes with different node locations or levels of sparsity. Additionally, they require significantly more memory resources than plain CNNs to store nodes' neighborhoods, which limits their deployment for complex meshes.

**Operator learning.** Operator learning is a burgeoning field in deep learning for physics that focuses on learning mappings between infinite-dimensional functions. Two prominent approaches are DeepONet (Lu et al., 2021) and Fourier Neural Operator (FNO; Li et al., 2021). DeepONet can query any coordinate in the domain for a value of the output function. However, the input function must be observed on a set of predefined locations, requiring the same observation grid for all observations, for training and testing. FNO is an instance of neural operators (Kovachki et al., 2021), a family of approaches that integrate kernels over the spatial domain. Since this operation can be expensive, FNO addresses the problem by employing the fast Fourier transform (FFT) to transform the inputs into the spectral domain. As a consequence it cannot be used with irregular grids. Li et al. (2022a) introduce an FNO extension to handle more flexible geometries, but it is tailored for design problems. To summarize, despite promising for several applications, current operator approaches still face limitations to extrapolate to new geometries; they do not adapt to changing observation grids or are limited to fixed observation locations. Recently, Li et al. (2023); Hao et al. (2023) explored transformer-based architectures as an alternative approach.

**Initial Value Problem**

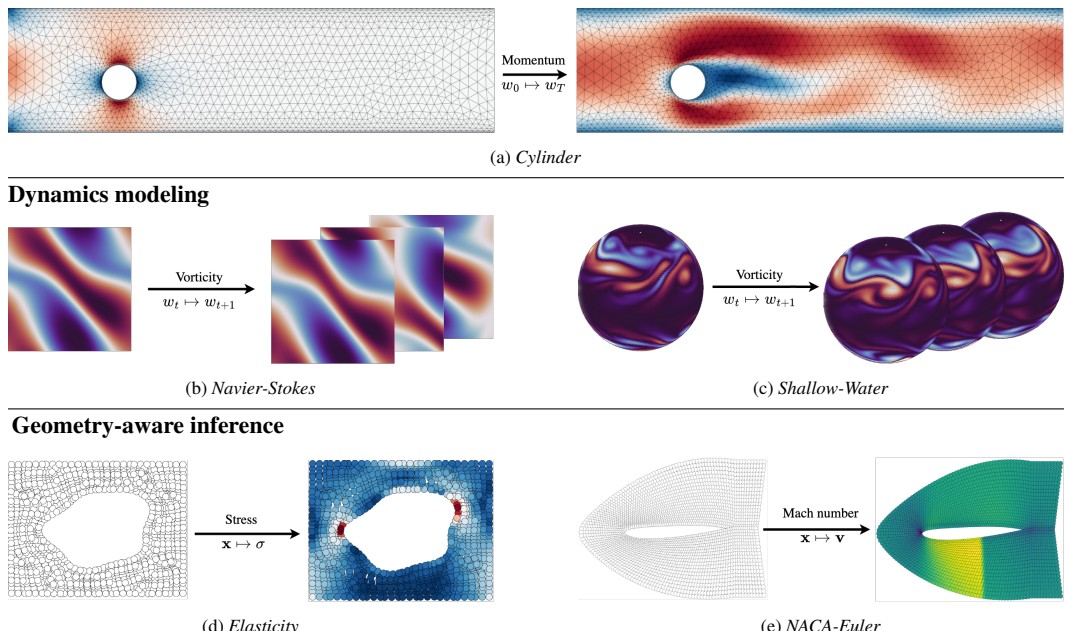

(a) *Cylinder*

**Dynamics modeling**

(b) *Navier-Stokes*

(c) *Shallow-Water*

**Geometry-aware inference**

(d) *Elasticity*

(e) *NACA-Euler*

Figure 1: Illustration of the problem classes addressed in this work: Initial Value Problem (IVP) (a), dynamic forecasting (b and c) and geometry-aware inference (d and e).

**Spatial INRs.** Spatial INRs are a class of coordinate-based neural networks that model data as the realization of an implicit function of a spatial location $x \in \Omega \mapsto f_\theta(x)$ (Tancik et al., 2020; Sitzmann et al., 2020b; Fathony et al., 2021; Lindell et al., 2022). An INR can be queried at any location, but encodes only one data sample or function. Previous works use meta-learning (Tancik et al., 2021; Sitzmann et al., 2020a), auto-encoders (Chen & Zhang, 2019; Mescheder et al., 2019), or modulation (Park et al., 2019; Dupont et al., 2022) to address this limitation by enabling an INR to decode various functions using per-sample parameters. INRs have started to gain traction in physics, where they have been successfully applied to spatio-temporal forecasting (Yin et al., 2022) and reduced-order modeling (Chen et al., 2022). The former work is probably the closest to ours but it is designed for forecasting and cannot handle the range of tasks that CORAL can address. Moreover, its computational cost is significantly higher than CORAL's, which limits its application in real-world problems. The work by Chen et al. (2022) aims to inform the INR with known PDEs, similar to PINNs, whereas our approach is entirely data-driven and without physical prior.

## 3 The CORAL Framework

In this section, we present the CORAL framework, a novel approach that employs an encode-process-decode structure to achieve the mapping between continuous functions. We first introduce the model and then the training procedure.

### 3.1 Problem Description

Let $\Omega \subset \mathbb{R}^d$ be a bounded open set of spatial coordinates. We assume the existence of a mapping $\mathcal{G}^*$ from one infinite-dimensional space $\mathcal{A} \subset L^2(\Omega, \mathbb{R}^{d_a})$ to another one $\mathcal{U} \subset L^2(\Omega, \mathbb{R}^{d_u})$, such that for any observed pairs $(a_i, u_i) \in \mathcal{A} \times \mathcal{U}$, $u_i = \mathcal{G}^*(a_i)$. We have $a_i \sim \nu_a$, $u_i \sim \nu_u$ where $\nu_a$ is a probability measure supported on $\mathcal{A}$ and $\nu_u$ the pushforward measure of $\nu_a$ by $\mathcal{G}^*$. We seek to approximate this operator by an i.i.d. collection of point-wise evaluations of input-output functions through a highly flexible formulation that can be adapted to multiple tasks. In this work, we target three different tasks as examples: • solving an initial value problem, i.e. mapping the initial condition $u_0 \doteq x \mapsto u(x, t = 0)$ to the solution at a predefined time $u_T \doteq x \mapsto u(x, t = T)$, • modeling the dynamics of a physical system over time $(u_t \to u_{t+\delta t})$ over a given forecasting horizon • or

prediction based on geometric configuration. At training time, we have access to $n_{tr}$ pairs of input and output functions $(a_i, u_i)_{i=1}^{n_{tr}}$ evaluated over a free-form spatial grid $\mathcal{X}_i$. We denote $a|_{\mathcal{X}_i} = (a(x))_{x \in \mathcal{X}_i}$ and $u|_{\mathcal{X}_i} = (u(x))_{x \in \mathcal{X}_i}$ the vectors of the function values over the sample grid. In the context of the initial value and geometry-aware problems, every sample is observed on a specific grid $\mathcal{X}_i$. For dynamics modeling, we use a unique grid $\mathcal{X}_{tr}$ for all the examples to train the model and another grid $\mathcal{X}_{te}$ for testing.

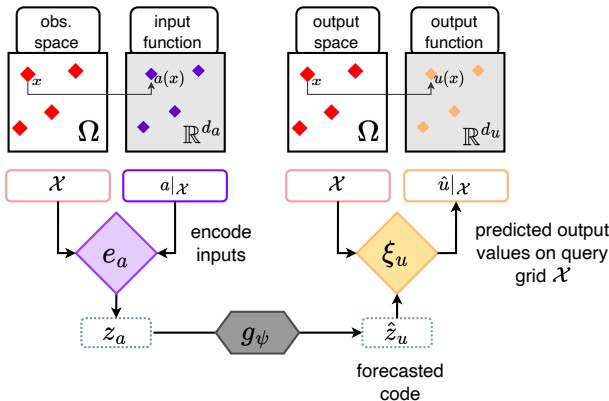

Figure 2: Inference for CORAL. First, the model embeds the input function $a$ without constraints on the locations of the observed sensors into an input latent code $z_a$, then infers the output latent code $\hat{z}_u$ and finally predicts the output value $\hat{u}(x)$ for any query coordinate $x \in \Omega$. For the grid $\mathcal{X}$, we use the vector notation $a|_{\mathcal{X}} = (a(x))_{x \in \mathcal{X}}$, $\hat{u}|_{\mathcal{X}} = (\hat{u}(x))_{x \in \mathcal{X}}$.

### 3.2 Model

CORAL makes use of two modulated INRs, $f_{\theta_a, \phi_a}$ and $f_{\theta_u, \phi_u}$, for respectively representing the input and output functions of an operator. While $\theta_a$ and $\theta_u$ denote shared INR parameters that contribute in representing all functions $a_i$ and $u_i$, the modulation parameters $\phi_{a_i}$ and $\phi_{u_i}$ are specific to each function $a_i$ and $u_i$. Given input/output INR functions representation, CORAL then learns a mapping between latent representations inferred from the two INRs' modulation spaces. The latent representations $z_{a_i}, z_{u_i}$ are low dimensional embeddings, capturing within a compact code information from the INRs' parameters. They are used as inputs to hypernetworks $h_a$ and $h_u$ to compute the modulation parameters $\phi_{a_i} = h_a(z_{a_i})$ and $\phi_{u_i} = h_u(z_{u_i})$. The weights of the input and output hypernetworks are respectively denoted $w_a$ and $w_u$.

CORAL proceeds in three steps: *encode*, to project the input data into the latent space; *process*, to perform transformations in the latent space; and *decode*, to project the code back to the output function space. First, the input function $a$ is encoded into the small input latent code $z_a$ using a spatial encoder $e_a : \mathcal{A} \mapsto \mathbb{R}^{d_z}$. Next, a parameterized model $g_\psi : \mathbb{R}^{d_z} \mapsto \mathbb{R}^{d_z}$ is used to infer an output latent code. Depending on the target task, $g_\psi$ can be as simple as a plain MLP or more complex as for example a neural ODE solver (as detailed later). Finally, the processed latent code is decoded into a spatial function using a decoder $\xi_u : \mathbb{R}^{d_z} \mapsto \mathcal{U}$. The resulting CORAL operator then writes as $\mathcal{G} = \xi_u \circ g_\psi \circ e_a$, as shown in Figure 2. The three steps are detailed below.

***Encode*** Given an input function $a_i$ and a learned shared parameter $\theta_a$, the encoding process provides a code $z_{a_i} = e_a(a_i)$. This code is computed by solving an inverse problem through a procedure known as *auto-decoding*, which proceeds as follows. We want to compress into a compact code $z_{a_i}$ the information required for reconstructing the original field $a_i$ through the input INR, i.e.: $\forall x \in \mathcal{X}_i, f_{\theta_a, \phi_{a_i}}(x) = \tilde{a}_i(x) \approx a_i(x)$ with $\phi_{a_i} = h_a(z_{a_i})$. See Figure 3a in Appendix B for details. The approximate solution to this inverse problem is computed as the solution $e_a(a_i) = z_{a_i}^{(K)}$ of a gradient descent optimization:

$$z_{a_i}^{(0)} = 0 \,; z_{a_i}^{(k+1)} = z_{a_i}^{(k)} - \alpha \nabla_{z_{a_i}^{(k)}} \mathcal{L}_{\mu_i}(f_{\theta_a, \phi_{a_i}^{(k)}}, a); \text{ with } \phi_{a_i}^{(k)} = h_a(z_{a_i}^{(k)}) \text{ for } 0 \leq k \leq K-1 \quad (1)$$

where $\alpha$ is the inner loop learning rate, $K$ the number of inner steps, and $\mathcal{L}_{\mu_i}(v, w) = \mathbb{E}_{x \sim \mu_i}[(v(x) - w(x))^2]$ for a measure $\mu_i$ over $\Omega$. Note that in practice, $\mu_i$ is defined through the observation grid

$\mathcal{X}_i$, $\mu_i(\cdot) = \sum_{x \in \mathcal{X}_i} \delta_x(\cdot)$ where $\delta_x(\cdot)$ is the Dirac measure. Since we can query the INRs anywhere within the domain, we can hence freely encode functions without mesh constraints. This is the essential part of the architecture that enables us to feed data defined on different grids to the model. We show the encoding flow in Appendix B, Figure 4.

**Process**    Once we obtain $z_{a_i}$, we can infer the latent output code $\hat{z}_{u_i} = g_\psi(z_{a_i})$. For simplification, we consider first that $g_\psi$ is implemented through an MLP with parameters $\psi$. For dynamics modeling, in Section 4.2, we will detail why and how to make use of a Neural ODE solver for $g_\psi$.

**Decode**    We decode $\hat{z}_{u_i}$ with the output hypernetwork $h_u$ and modulated INR and denote $\xi_u$ the mapping that associates to code $\hat{z}_{u_i}$ the function $f_{\theta_u, \hat{\phi}_{u_i}}$, where $\hat{\phi}_{u_i} = h_u(\hat{z}_{u_i})$. Since $f_{\theta_u, \hat{\phi}_{u_i}}$ is an INR, i.e. a function of spatial coordinates, it can be freely queried at any point within the domain. We thus have $\forall x \in \Omega, \hat{u}_i(x) = \xi_u(\hat{z}_{u_i})(x) = f_{\theta_u, \hat{\phi}_{u_i}}(x)$. See Figure 3b in Appendix B for details.

During training, we will need to learn to reconstruct the input and output functions $a_i$ and $u_i$. This requires training a mapping associating an input code to the corresponding input function $\xi_a : \mathbb{R}^{d_z} \mapsto \mathcal{A}$ and a mapping associating a function to its code in the output space $e_u : \mathcal{U} \mapsto \mathbb{R}^{d_z}$, even though they are not used during inference.

### 3.3    Practical implementation: decoding by INR Modulation

We choose SIREN (Sitzmann et al., 2020b) – a state-of-the-art coordinate-based network – as the INR backbone of our framework. SIREN is a neural network that uses sine activations with a specific initialization scheme (Appendix B).

$$f_\theta(x) = \boldsymbol{W}_L\big(\sigma_{L-1} \circ \sigma_{L-2} \circ \cdots \circ \sigma_0(x)\big) + \boldsymbol{b}_L, \text{ with } \sigma_i(\eta_i) = \sin\big(\omega_0(\boldsymbol{W}_i\eta_i + \boldsymbol{b}_i)\big) \qquad (2)$$

where $\eta_0 = x$ and $(\eta_i)_{i \geq 1}$ are the hidden activations throughout the network. $\omega_0 \in \mathbb{R}_+^*$ is a hyperparameter that controls the frequency bandwidth of the network, $\boldsymbol{W}$ and $\boldsymbol{b}$ are the network weights and biases. We implement shift modulations (Perez et al., 2018) to have a small modulation space and reduce the computational cost of the overall architecture. This yields the modulated SIREN:

$$f_{\theta,\phi}(x) = \boldsymbol{W}_L\big(\sigma_{L-1} \circ \sigma_{L-2} \circ \cdots \circ \sigma_0(x)\big) + \boldsymbol{b}_L, \text{ with } \sigma_i(\eta_i) = \sin\big(\omega_0(\boldsymbol{W}_i\eta_i + \boldsymbol{b}_i + \phi_i)\big) \quad (3)$$

with shared parameters $\theta = (\boldsymbol{W}_i, \boldsymbol{b}_i)_{i=0}^L$ and example associated modulations $\phi = (\phi_i)_{i=0}^{L-1}$. We compute the modulations $\phi$ from $z$ with a linear hypernetwork , i.e. for $0 \leq i \leq L - 1$, $\phi_i = \boldsymbol{V}_i z + \boldsymbol{c}_i$. The weights $\boldsymbol{V}_i$ and $\boldsymbol{c}_i$ constitute the parameters of the hypernetwork $w = (\boldsymbol{V}_i, \boldsymbol{c}_i)_{i=0}^{L-1}$. This implementation is similar to that of Dupont et al. (2022), which use a modulated SIREN for representing their modalities.

### 3.4    Training

We implement a two-step training procedure that first learns the modulated INR parameters, before training the forecast model $g_\psi$. It is very stable and much faster than end-to-end training while providing similar performance: once the input and output INRs have been fitted, the training of $g_\psi$ is performed in the small dimensional modulated INR $z$-code space. Formally, the optimization problem is defined as:

$$\arg\min_\psi \mathbb{E}_{a,u \sim \nu_a, \nu_u} \|g_\psi(\tilde{e}_a(a)) - \tilde{e}_u(u)\|^2$$

$$\text{s.t. } \tilde{e}_a = \arg\min_{\xi_a, e_a} \mathbb{E}_{a \sim \nu_a} \mathcal{L}(\xi_a \circ e_a(a), a) \qquad (4)$$

$$\text{and } \tilde{e}_u = \arg\min_{\xi_u, e_u} \mathbb{E}_{u \sim \nu_u} \mathcal{L}(\xi_u \circ e_u(u), u)$$

Note that functions $(e_u, \xi_u)$ and $(e_a, \xi_a)$ are parameterized respectively by the weights $(\theta_u, w_u)$ and $(\theta_a, w_a)$, of the INRs and of the hypernetworks. In Equation (4), we used the $(e_u, \xi_u)$ & $(e_a, \xi_a)$ description for clarity, but as they are functions of $(\theta_u, w_u)$ & $(\theta_a, w_a)$, optimization is tackled on the latter parameters. We outline the training pipeline in Appendix B, Figure 5. During training, we

constrain $e_u, e_a$ to take only a few steps of gradient descent to facilitate the processor task. This regularization prevents the architecture from memorizing the training set into the individual codes and facilitates the auto-decoding optimization process for new inputs. In order to obtain a network that is capable of quickly encoding new physical inputs, we employ a second-order meta-learning training algorithm based on CAVIA (Zintgraf et al., 2019). Compared to a first-order scheme such as Reptile (Nichol et al., 2018), the outer loop back-propagates the gradient through the $K$ inner steps, consuming more memory as we need to compute gradients of gradients but yielding higher reconstruction results with the modulated SIREN. We experimentally found that using 3 inner-steps for training, or testing, was sufficient to obtain very low reconstruction errors for most applications.

## 4 Experiments

To demonstrate the versatility of our model, we conducted experiments on three distinct tasks (Figure 1): (i) solving an initial value problem (Section 4.1), (ii) modeling the dynamics of a physical system (Section 4.2), and (iii) learning to infer the steady state of a system based on the domain geometry (Section 4.3) plus an associated design problem in Appendix D. Since each task corresponds to a different scenario, we utilized task-specific datasets and employed different baselines for each task. This approach was necessary because existing baselines typically focus on specific tasks and do not cover the full range of problems addressed in our study, unlike CORAL. We provide below an introduction to the datasets, evaluation protocols, and baselines for each task setting. All experiments were conducted on a single GPU: NVIDIA RTX A5000 with 25 Go. Code will be made available.

### 4.1 Initial Value Problem

An IVP is specified by an initial condition (here the input function providing the state variables at $t = 0$) and a target function figuring the state variables value at a given time $T$. Solving an IVP is a direct application of the CORAL framework introduced in Section 3.2.

**Datasets** We benchmark our model on two problems with non-convex domains proposed in Pfaff et al. (2021). In both cases, the fluid evolves in a domain – which includes an obstacle – that is more densely discretized near the boundary conditions (BC). The boundary conditions are provided by the mesh definition, and the models are trained on multiple obstacles and evaluated at test time on similar but different obstacles. • **Cylinder** simulates the flow of water around a cylinder on a fixed 2D Eulerian mesh, and is characteristic of *incompressible* fluids. For each node $j$ we have access to the node position $x^{(j)}$, the momentum $w(x^{(j)})$ and the pressure $p(x^{(j)})$. We seek to learn the mapping $(x, w_0(x), p_0(x))_{x \in \mathcal{X}} \rightarrow (w_T(x), p_T(x))_{x \in \mathcal{X}}$. • **Airfoil** simulates the aerodynamics around the cross-section of an airfoil wing, and is an important use-case for *compressible* fluids. In this dataset, we have in addition for each node $j$ the fluid density $\rho(x^{(j)})$, and we seek to learn the mapping $(x, w_0(x), p_0(x), \rho_0(x))_{x \in \mathcal{X}} \rightarrow (w_T(x), p_T(x), \rho_T(x))_{x \in \mathcal{X}}$. For both datasets, each example is associated to a mesh and the meshes are different for each example. For *Airfoil* the average number of nodes per mesh is 5233 and for *Cylinder* 1885.

**Evaluation protocols** Training is performed using all the mesh points associated to an example. For testing we evaluate the following two settings. • **Full**, we validate that the trained model generalizes well to new examples using all the mesh location points of these examples. • **Sparse** We assess the capability of our model to generalize on sparse meshes: the original input mesh is down-sampled by randomly selecting 20% of its nodes. We use a train, validation, test split of 1000 / 100 / 100 samples for all the evaluations.

**Baselines** We compare our model to • **NodeMLP**, a FeedForward Neural Network that ignores the node neighbors and only learns a local mapping • **GraphSAGE** (Hamilton et al., 2017), a popular GNN architecture that uses SAGE convolutions • **MP-PDE** (Brandstetter et al., 2022b), a message passing GNN that builds on (Pfaff et al., 2021) for solving PDEs.

**Results.** We show in Table 1 the performance on the test sets for the two datasets and for both evaluation settings. Overall, CORAL is on par with the best models for this task. For the *Full* setting, it is best on *Cylinder* and second on *Airfoil* behind MP-PDE. However, for the *sparse* protocol, it can infer the values on the full mesh with the lowest error compared to all other models. Note that this

second setting is more challenging for *Cylinder* than for *Airfoil* given their respective average mesh size. This suggests that the interpolation of the model outputs is more robust on the *Airfoil* dataset, and explains why the performance of NodeMLP remains stable between the two settings. While MP-PDE is close to CORAL in the *sparse* setting, GraphSAGE fails to generalize, obtaining worse predictions than the local model. This is because the model aggregates neighborhood information regardless of the distance between nodes, while MP-PDE does consider node distance and difference between features.

Table 1: **Initial Value Problem** - Test results. MSE on normalized data.

| Model | Cylinder | | Airfoil | |
|---|---|---|---|---|
| | *Full* | *Sparse* | *Full* | *Sparse* |
| NodeMLP | 1.48e-1 $\pm$ 2.00e-3 | 2.29e-1 $\pm$ 3.06e-3 | 2.88e-1 $\pm$ 1.08e-2 | 2.83e-1 $\pm$ 2.12e-3 |
| GraphSAGE | 7.40e-2 $\pm$ 2.22e-3 | 2.66e-1 $\pm$ 5.03e-3 | 2.47e-1 $\pm$ 7.23e-3 | 5.55e-1 $\pm$ 5.54e-2 |
| MP-PDE | 8,72e-2 $\pm$ 4.65e-3 | 1.84e-1 $\pm$ 4.58e-3 | **1.97e-1 $\pm$ 1.34e-2** | 3.07e-1 $\pm$ 2.56e-2 |
| CORAL | **7.03e-2 $\pm$ 5.96e-3** | **1.70e-1 $\pm$ 2.53e-2** | 2.40e-1 $\pm$ 4.36e-3 | **2.43e-1 $\pm$ 4.14e-3** |

## 4.2 Dynamics Modeling

For the IVP problem, in section 4.1, the objective was to infer directly the state of the system at a given time $T$ given an initial condition (IC). We can extend this idea to model the dynamics of a physical system over time, so as to forecast state values over a given horizon. We have developed an autoregressive approach operating on the latent code space for this problem. Let us denote $(u_0, u_{\delta t}, ..., u_{L\delta t})$ a target sequence of observed functions of size $L + 1$. Our objective will be to predict the functions $u_{k\delta t}, k = 1, ..., L$, starting from an initial condition $u_0$. For that we will encode $z_0 = e(u_0)$, then predict sequentially the latent codes $z_{k\delta t}, k = 1, ..., L$ using the processor in an auto regressive manner, and decode the successive values to get the predicted $\hat{u}_{k\delta t}, k = 1, ..., L$ at the successive time steps.

### 4.2.1 Implementation with Neural ODE

The autoregressive processor is implemented by a Neural ODE solver operating in the latent $z$-code space. Compared to the plain MLP implementation used for the IVP task, this provides both a natural autoregressive formulation, and overall, an increased flexibility by allowing to forecast at any time in a sequence, including different time steps or irregular time steps. Starting from any latent state $z_t$, a neural solver predicts state $z_{t+\tau}$ as $z_{t+\tau} = z_t + \int_t^{t+\tau} \zeta_\psi(z_s)ds$ with $\zeta_\psi$ a neural network with parameters to be learned, for any time step $\tau$. The autoregressive setting directly follows from this formulation. Starting from $z_0$, and specifying a series of forecast time steps $k\delta t$ for $k = 1, ..., L$, the solver call $NODESolve(\zeta_\psi, z_0, \{k\delta t\}_{k=1,...,L})$ will compute predictions $z_{k\delta t}, k = 1, ..., L$ autoregressively, i.e. using $z_{k\delta t}$ as a starting point for computing $z_{(k+1)\delta t}$. In our experiments we have used a fourth-order Runge-Kutta scheme (RK4) for solving the integral term. Using notations from Section 3.2, the predicted field at time step $k$ can be obtained as $\hat{u}_{k\delta t} = \xi \circ g_\psi^k(e(u_0)))$ with $g_\psi^k$ indicating $k$ successive applications of processor $g_\psi$. Note that when solving the IVP problem from Section 4.1, two INRs are used, one for encoding the input function and one for the output function; here a single modulated INR $f_{\theta,\phi}$ is used to represent a physical quantity throughout the sequence at any time. $\theta$ is then shared by all the elements of a sequence and $\phi$ is computed by a hypernetwork to produce a function specific code.

We use the two-step training procedure from Section 3.4, i.e. first the INR is trained to auto-decode the states of each training trajectory, and then the processor operating over the codes is learned through a Neural ODE solver according to Equation (5). The two training steps are separated and the codes are kept fixed during the second step. This allows for a fast training as the Neural ODE solver operates on the low dimensional code embedding space.

$$\arg\min_\psi \mathbb{E}_{u\sim\nu_u, t\sim\mathcal{U}(0,T]} \|g_\psi(\tilde{e}_u(u_0), t) - \tilde{e}_u(u_t)\|^2 \tag{5}$$

#### 4.2.2 Experiment details

**Datasets** We consider two fluid dynamics equations for generating the datasets and refer the reader to Appendix A for additional details. • **2D-Navier-Stokes equation** (*Navier-Stokes*) for a viscous, incompressible fluid in vorticity form on the unit torus: $\frac{\partial w}{\partial t} + u \cdot \nabla w = \nu \Delta w + f$, $\nabla u = 0$ for $x \in \Omega, t > 0$, where $\nu = 10^{-3}$ is the viscosity coefficient. The train and test sets are composed of 256 and 16 trajectories respectively where we observe the vorticity field for 40 timestamps. The original spatial resolution is $256 \times 256$ and we sub-sample the data to obtain frames of size $64 \times 64$. • **3D-Spherical Shallow-Water equation** (*Shallow-Water*) can be used as an approximation to a flow on the earth's surface. The data consists of the vorticity $w$, and height $h$ of the fluid. The train and test sets are composed respectively of 16 and 2 long trajectories, where we observe the vorticity and height fields for 160 timestamps. The original spatial resolution is 128 (lat) $\times$ 256 (long), which we sub-sample to obtain frames of shape $64 \times 128$. We model the dynamics with the complete state $(h, w)$. Each trajectory, for both both datasets and for train and test is generated from a different initial condition (IC).

**Setting** We evaluate the ability of the model to generalize in space and time. • **Temporal extrapolation:** For both datasets, we consider sub-trajectories of 40 timestamps that we split in two equal parts of size 20, with the first half denoted *In-t* and the second one *Out-t*. The training-*In-t* set is used to train the models at forecasting the horizon $t = 1$ to $t = 19$. At test time, we unroll the dynamics from a new IC until $t = 39$. Evaluation in the horizon *In-t* assesses CORAL's capacity to forecast within the training horizon. *Out-t* allows evaluation beyond *In-t*, from $t = 20$ to $t = 39$. • **Varying sub-sampling:** We randomly sub-sample $\pi$ percent of a regular mesh to obtain the train grid $\mathcal{X}_{tr}$, and a second test grid $\mathcal{X}_{te}$, that are shared across trajectories. The train and test grids are different, but have the same level of sparsity. • **Up-sampling:** We also evaluate the up-sampling capabilities of CORAL in Appendix C. In these experiments, we trained the model on a sparse, low-resolution grid and evaluate its performance on high resolution-grids.

**Baselines** To assess the performance of CORAL, we implement several baselines: two operator learning models, one mesh-based network and one coordinate-based method. • **DeepONet** (Lu et al., 2021): we train DeepONet in an auto-regressive manner with time removed from the trunk net's input. • **FNO** (Li et al., 2021): we use an auto-regressive version of the Fourier Neural Operator. • **MP-PDE** (Brandstetter et al., 2022b) : we use MP-PDE as the irregular mesh-based baseline. We fix MP-PDE's temporal bundling to 1, and train the model with the push-forward trick. • **DINo** (Yin et al., 2022) : We finally compare CORAL with DINo, an INR-based model designed for dynamics modeling.

#### 4.2.3 Results

Table 2: **Temporal Extrapolation** - Test results. Metrics in MSE.

| $\mathcal{X}_{tr} \downarrow \mathcal{X}_{te}$ | dataset $\rightarrow$ | Navier-Stokes | | Shallow-Water | |
| --- | --- | --- | --- | --- | --- |
| | | *In-t* | *Out-t* | *In-t* | *Out-t* |
| $\pi = 100\%$ regular grid | DeepONet | 4.72e-2 $\pm$ 2.84e-2 | 9.58e-2 $\pm$ 1.83e-2 | 6.54e-3 $\pm$ 4.94e-4 | 8.93e-3 $\pm$ 9.42e-5 |
| | FNO | 5.68e-4 $\pm$ 7.62e-5 | 8.95e-3 $\pm$ 1.50e-3 | 3.20e-5 $\pm$ 2.51e-5 | **1.17e-4 $\pm$ 3.01e-5** |
| | MP-PDE | 4.39e-4 $\pm$ 8.78e-5 | 4.46e-3 $\pm$ 1.28e-3 | 9.37e-5 $\pm$ 5.56e-6 | 1.53e-3 $\pm$ 2.62e-4 |
| | DINo | 1.27e-3 $\pm$ 2.22e-5 | 1.11e-2 $\pm$ 2.28e-3 | 4.48e-5 $\pm$ 2.74e-6 | 2.63e-3 $\pm$ 1.36e-4 |
| | CORAL | **1.86e-4 $\pm$ 1.44e-5** | **1.02e-3 $\pm$ 8.62e-5** | **3.44e-6 $\pm$ 4.01e-7** | 4.82e-4 $\pm$ 5.16e-5 |
| $\pi = 20\%$ irregular grid | DeepONet | 8.37e-1 $\pm$ 2.07e-2 | 7.80e-1 $\pm$ 2.36e-2 | 1.05e-2 $\pm$ 5.01e-4 | 1.09e-2 $\pm$ 6.16e-4 |
| | FNO + lin. int. | 3.97e-3 $\pm$ 8.03e-4 | 9.92e-3 $\pm$ 2.36e-3 | n.a. | n.a. |
| | MP-PDE | 3,98e-2 $\pm$ 1,69e-2 | 1,31e-1 $\pm$ 5,34e-2 | 5.28e-3 $\pm$ 5.25e-4 | 2.56e-2 $\pm$ 8.23e-3 |
| | DINo | **9.99e-4 $\pm$ 6.71e-3** | 8.27e-3 $\pm$ 5.61e-3 | 2.20e-3 $\pm$ 1.06e-4 | 4.94e-3 $\pm$ 1.92e-4 |
| | CORAL | 2.18e-3 $\pm$ 6.88e-4 | **6.67e-3 $\pm$ 2.01e-3** | **1.41e-3 $\pm$ 1.39e-4** | **2.11e-3 $\pm$ 5.58e-5** |
| $\pi = 5\%$ irregular grid | DeepONet | 7.86e-1 $\pm$ 5.48e-2 | 7.48e-1 $\pm$ 2.76e-2 | 1.11e-2 $\pm$ 6.94e-4 | **1.12e-2 $\pm$ 7.79e-4** |
| | FNO + lin. int. | 3.87e-2 $\pm$ 1.44e-2 | 5.19e-2 $\pm$ 1.10e-2 | n.a. | n.a. |
| | MP-PDE | 1.92e-1 $\pm$ 9.27e-2 | 4.73e-1 $\pm$ 2.17e-1 | 1.10e-2 $\pm$ 4.23e-3 | 4.94e-2 $\pm$ 2.36e-2 |
| | DINo | 8.65e-2 $\pm$ 1.16e-2 | 9.36e-2 $\pm$ 9.34e-3 | **1.22e-3 $\pm$ 2.05e-4** | 1.52e-2 $\pm$ 3.74e-4 |
| | CORAL | **2.44e-2 $\pm$ 1.96e-2** | **4.57e-2 $\pm$ 1.78e-2** | 8.77e-3 $\pm$ 7.20e-4 | 1.29e-2 $\pm$ 1.92e-3 |

Table 2 details the performance of the different models in a combined temporal and spatial evaluation setting. ● **General remarks:** CORAL demonstrates strong performance across all scenarios for both datasets. Only DINo exhibits similar properties, i.e., stability across spatial subsamplings and extrapolation horizon. We observe that all models performance degrade with lower sampling ratio. Also, as the models have been trained only on *In-t* horizon, error accumulates over time and thus leads to lower performance for *Out-t* evaluation. ● **Analysis per model:** Although achieving strong performance on some specific scenarios, DeepONet, FNO and MP-PDE results are dependent of the training grid, geometries or number of points. FNO, can only be trained and evaluated on regular grids while DeepONet is not designed to be evaluated on a different grid in the branch net. MP-PDE achieves strong performance with enough sample positions, e.g. full grids here, but struggles to compete on irregular grids scenarios in Navier-Stokes. ● **Inference Time:** We report in Appendix C, the inference time of the baselines considered. Despite operator methods have better inference time, CORAL is faster than mesh-free methods like DINo and MP-PDE. ● **Generalization across samplings:** Coordinate-based methods demonstrate robustness when it comes to changes in spatial resolution. In contrast, MP-PDE model exhibits strong overfitting to the training grid, resulting in a decline in performance. Although MP-PDE and DINo may outperform CORAL in some settings, when changing the grid, CORAL remains stable and outperforms the other models. See Appendix C for details.

## 4.3 Geometry-aware inference

In this section, we wish to infer the steady state of a system from its domain geometry, all other parameters being equal. The domain geometry is partially observed from the data in the form of point clouds or of a structured mesh $\mathcal{X}_i \subset \Omega_i$. The position of the nodes depends on the particular object shape. Each mesh $\mathcal{X}_i$ is obtained by deforming a reference grid $\mathcal{X}$ to adjust to the shape of the sample object. This grid deformation is the input function of the operator learning setting, while the output function is the physical quantity $u_i$ over the domain $\Omega_i$. The task objective is to train a model so as to generalize to new geometries, e.g. a new airfoil shape. Once a surrogate model has been trained to learn the influence of the domain geometry on the steady state solution, it can be used to quickly evaluate a new design and to solve inverse design problems (details in Appendix D).

**Datasets.** We used datasets generated from three different equations by Li et al. (2022a) and provide more details in Appendix A. ● **Euler equation** (*NACA-Euler*) for a transonic flow over a NACA-airfoil. The measured quantity at each node is the Mach number. ● **Navier-Stokes Equation** (*Pipe*) for an incompressible flow in a pipe, expressed in velocity form. The measured quantity at each node is the horizontal velocity. ● **Hyper-elastic material** (*Elasticity*). Each sample represents a solid body with a void in the center of arbitrary shape, on which a tension is applied at the top. The material is the incompressible Rivlin-Saunders material and the measured quantity is the stress value. We use 1000 samples for training and 200 for test with all datasets.

**Baselines** We use ● **Geo-FNO** (Li et al., 2022a) and ● **Factorized-FNO** (Tran et al., 2023) two SOTA models as the main baselines. We also compare our model to regular-grid methods such as ● **FNO** (Li et al., 2021) and ● **UNet** (Ronneberger et al., 2015), for which we first interpolate the input.

Table 3: **Geometric aware inference** - Test results. Relative L2 error.

| Model | NACA-Euler | Elasticity | Pipe |
|---|---|---|---|
| FNO | 3.85e-2 ± 3.15e-3 | 4.95e-2 ± 1.21e-3 | 1.53e-2 ± 8.19e-3 |
| UNet | 5.05e-2 ± 1.25e-3 | 5.34e-2 ± 2.89e-4 | 2.98e-2 ± 1.08e-2 |
| Geo-FNO | 1.58e-2 ± 1.77e-3 | 3.41e-2 ± 1.93e-2 | **6.59e-3 ± 4.67e-4** |
| Factorized-FNO | 6.20e-3 ± 3.00e-4 | 1.96e-2 ± 2.00e-2 | 7.33e-3 ± 4.66e-4 |
| CORAL | **5.90e-3 ± 1.00e-4** | **1.67e-2 ± 4.18e-4** | 1,20e-2 ± 8.74e-4 |

**Results** In Table 3 we can see that CORAL achieves state-of-the-art results on *Airfoil* and *Elasticity*, with the lowest relative error among all models. It is slightly below Factorized-FNO and Geo-FNO on *Pipe*. One possible cause is that this dataset exhibits high frequency only along the vertical dimension, while SIREN might be better suited for isotropic frequencies. Through additional experiments, we demonstrate in Appendix D, how CORAL can also be used for solving an inverse problem

corresponding to a design task: optimize the airfoil geometry to minimize the drag over lift ratio. This additional task further highlights the versatility of this model.

## 5 Discussion and limitations

Although a versatile model, CORAL inherits the limitations of INRs concerning the training time and representation power. It is then faster to train than GNNs, but slower than operators such as FNO, DeepONet and of course CNNs which might limit large scale deployments. Also some physical phenomena might not be represented via INRs. Although this is beyond the scope of this paper, it remains to evaluate the methods on large size practical problems. An interesting direction for future work would be to derive an efficient spatial latent representation for INRs, taking inspiration from grid-based representation for INRs (Takikawa et al. (2022), Müller et al. (2022), Saragadam et al. (2022)). Another avenue would be to leverage clifford layers (Brandstetter et al., 2022a) to model interactions between physical fields.

## 6 Conclusion

We have presented CORAL, a novel approach for Operator Learning that removes constraints on the input-output mesh. CORAL offers the flexibility to handle spatial sampling or geometry variations, making it applicable to a wide range of scenarios. Through comprehensive evaluations on diverse tasks, we have demonstrated that it consistently achieves state-of-the-art or competitive results compared to baseline methods. By leveraging compact latent codes to represent functions, it enables efficient and fast inference within a condensed representation space.

## Acknowledgements

We acknowledge the financial support provided by DL4CLIM (ANR-19-CHIA-0018-01) and DEEP-NUM (ANR-21-CE23-0017-02) ANR projects, as well as the program France 2030, project 22-PETA-0002. This project was provided with computer and storage resources by GENCI at IDRIS thanks to the grant AD011013522 on the supercomputer Jean Zay's V100 partition.

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

# A  Dataset Details

## A.1  Initial Value Problem

We use the datasets from Pfaff et al. (2021), and take the first and last frames of each trajectory as the input and output data for the initial value problem.

**Cylinder**   The dataset includes computational fluid dynamics (CFD) simulations of the flow around a cylinder, governed by the incompressible Navier-Stokes equation. These simulations were generated using COMSOL software, employing an irregular 2D-triangular mesh. The trajectory consists of 600 timestamps, with a time interval of $\Delta t = 0.01s$ between each timestamp.

**Airfoil**   The dataset contains CFD simulations of the flow around an airfoil, following the compressible Navier-Stokes equation. These simulations were conducted using SU2 software, using an irregular 2D-triangular mesh. The trajectory encompasses 600 timestamps, with a time interval of $\Delta t = 0.008s$ between each timestamp.

## A.2  Dynamics Modeling

**2D-Navier-Stokes** (*Navier-Stokes*)   We consider the 2D Navier-Stokes equation as presented in Li et al. (2021); Yin et al. (2022). This equation models the dynamics of an incompressible fluid on a rectangular domain $\Omega = [-1, 1]^2$. The PDE writes as :

$$\frac{\partial w(x,t)}{\partial t} = -u(x,t)\nabla w(x,t) + \nu \Delta w(x,t) + f, x \in [-1,1]^2, t \in [0,T] \tag{6}$$

$$w(x,t) = \nabla \times u(x,t), x \in [-1,1]^2, t \in [0,T] \tag{7}$$

$$\nabla u(x,t) = 0, x \in [-1,1]^2, t \in [0,T] \tag{8}$$

where $u$ is the velocity, $w$ the vorticity. $\nu$ is the fluid viscosity, and $f$ is the forcing term, given by:

$$f(x_1, x_2) = 0.1\left(\sin(2\pi(x_1 + x_2)) + \cos(2\pi(x_1 + x_2))\right), \forall x \in \Omega \tag{9}$$

For this problem, we consider periodic boundary conditions.

By sampling initial conditions as in Li et al. (2021), we generated different trajectories on a $256 \times 256$ regular spatial grid and with a time resolution $\delta t = 1$. We retain the trajectory starting from the 20th timestep so that the dynamics is sufficiently expressed. The final trajectories contains 40 snapshots at time $t = 20, 21, \cdots, 59$. As explained in section 4, we divide these long trajectories into 2 parts : the 20 first frames are used during the training phase and are denoted as *In-t* throughout this paper. The 20 last timesteps are reserved for evaluating the extrapolation capabilities of the models and are the *Out-t* part of the trajectories. In total, we collected 256 trajectories for training, and 16 for evaluation.

**3D-Spherical Shallow-Water** (*Shallow-Water*).   We consider the shallow-water equation on a sphere describing the movements of the Earth's atmosphere:

$$\frac{\mathrm{d}u}{\mathrm{d}t} = -f \cdot k \times u - g\nabla h + \nu \Delta u \tag{10}$$

$$\frac{\mathrm{d}h}{\mathrm{d}t} = -h\nabla \cdot u + \nu \Delta h \tag{11}$$

where $\frac{\mathrm{d}}{\mathrm{d}t}$ is the material derivative, $k$ is the unit vector orthogonal to the spherical surface, $u$ is the velocity field tangent to the surface of the sphere, which can be transformed into the vorticity $w = \nabla \times u$, $h$ is the height of the sphere. We generate the data with the *Dedalus* software (Burns et al., 2020), following the setting described in Yin et al. (2022), where a symmetric phenomena can be seen for both northern and southern hemisphere. The initial zonal velocity $u_0$ contains two non-null symmetric bands in the both hemispheres, which are parallel to the circles of latitude. At each latitude and longitude $\phi, \theta \in [-\frac{\pi}{2}, \frac{\pi}{2}] \times [-\pi, \pi]$:

$$u_0(\phi, \theta) = \begin{cases} \left(\frac{u_{max}}{e_n} \exp\left(\frac{1}{(\phi - \phi_0)(\phi - \phi_1)}\right), 0\right) & \text{if } \phi \in (\phi_0, \phi_1), \\ \left(\frac{u_{max}}{e_n} \exp\left(\frac{1}{(\phi + \phi_0)(\phi + \phi_1)}\right), 0\right) & \text{if } \phi \in (-\phi_1, -\phi_0), \\ (0, 0) & \text{otherwise.} \end{cases} \tag{12}$$

where $u_{max}$ is the maximum velocity, $\phi_0 = \frac{\pi}{7}$, $\phi_1 = \frac{\pi}{2} - \phi_0$, and $e_n = \exp(-\frac{4}{(\phi_1 - \phi_0)^2})$. The water height $h_0$ is initialized by solving a boundary value conditioned problem as in Galewsky et al. (2004) which is perturbed by adding $h_0'$ to $h_0$:

$$h_0'(\phi, \theta) = \hat{h}\cos(\phi)\exp\left(-\left(\frac{\theta}{\alpha}\right)^2\right)\left[\exp\left(-\left(\frac{\phi_2 - \phi}{\beta}\right)^2\right) + \exp\left(-\left(\frac{\phi_2 + \phi}{\beta}\right)^2\right)\right]. \quad (13)$$

where $\phi_2 = \frac{\pi}{4}$, $\hat{h} = 120\text{m}$, $\alpha = \frac{1}{3}$ and $\beta = \frac{1}{15}$ are constants defined in Galewsky et al. (2004). We simulated the phenomenon using Dedalus Burns et al. (2020) on a latitude-longitude grid (lat-lon). The original grid size was 128 (lat) $\times$ 256 (lon), which we downsampled to obtain grids of size $64 \times 128$. To generate trajectories, we sampled $u_{max}$ from a uniform distribution $\mathcal{U}(60, 80)$. Snapshots were captured every hour over a duration of 320 hours, resulting in trajectories with 320 timestamps. We created 16 trajectories for the training set and 2 trajectories for the test set. However, since the dynamical phenomena in the initial timestamps were less significant, we only considered the last 160 snapshots. Each long trajectory is then sliced into sub-trajectories of 40 timestamps each. As a result, the training set contains 64 trajectories, while the test set contains 8 trajectories. It is worth noting that the data was also scaled to a reasonable range: the height $h$ was scaled by a factor of $3 \times 10^3$, and the vorticity $w$ was scaled by a factor of 2.

### A.3  Geometric aware inference

We use the datasets provided by Li et al. (2022a) and adopt the original authors' train/test split for our experiments.

**Euler's Equation** (*Naca-Euler*).    We consider the transonic flow over an airfoil, where the governing equation is Euler equation, as follows:

$$\frac{\partial \rho_f}{\partial t} + \nabla \cdot (\rho_f u) = 0, \frac{\partial \rho_f u}{\partial t} + \nabla \cdot (\rho_f u \otimes u + p\mathbb{I}) = 0, \frac{\partial E}{\partial t} + \nabla \cdot ((E + p)u) = 0, \quad (14)$$

where $\rho_f$ is the fluid density, $u$ is the velocity vector, $p$ is the pressure, and $E$ is the total energy. The viscous effect is ignored. The far-field boundary condition is $\rho_\infty = 1$, $p_\infty = 1.0$, $M_\infty = 0.8$, $AoA = 0$, where $M_\infty$ is the Mach number and $AoA$ is the angle of attack. At the airfoil, a no-penetration condition is imposed. The shape parameterization of the airfoil follows the design element approach. The initial NACA-0012 shape is mapped onto a "cubic" design element with 8 control nodes, and the initial shape is morphed to a different one following the displacement field of the control nodes of the design element. The displacements of control nodes are restricted to the vertical direction only, with prior $d \sim \mathcal{U}[-0.05, 0.05]$.

We have access to 1000 training data and 200 test data, generated with a second-order implicit finite volume solver. The C-grid mesh with about $(200 \times 50)$ quadrilateral elements is used, and the mesh is adapted near the airfoil but not the shock. The mesh point locations and Mach number on these mesh points are used as input and output data.

**Hyper-elastic material** (*Elasticity*).    The governing equation of a solid body can be written as

$$\rho_s \frac{\partial^2 u}{\partial t^2} + \nabla \cdot \sigma = 0$$

where $\rho_s$ is the mass density, $u$ is the displacement vector, and $\sigma$ is the stress tensor. Constitutive models, which relate the strain tensor $\varepsilon$ to the stress tensor, are required to close the system. We consider the unit cell problem $\Omega = [0, 1] \times [0, 1]$ with an arbitrary shape void at the center, which is depicted in Figure 2(a). The prior of the void radius is $r = 0.2 + 0.2$ with $\tilde{r} \sim \mathcal{N}(0, 42(-\nabla + 32)^{-1})$, $1 + \exp(\tilde{r})$, which embeds the constraint $0.2 \leq r \leq 0.4$. The unit cell is clamped on the bottom edges and tension traction $t = [0, 100]$ is applied on the top edge. The material is the incompressible Rivlin-Saunders material with energy density function parameters $C_1 = 1.863 \times 10^5$ and $C_1 = 9.79 \times 10^3$. The data was generated with a finite element solver with about 100 quadratic quadrilateral elements. The inputs $a$ are given as point clouds with a size around 1000. The target output is stress.

**Navier-Stokes Equation** (*Pipe*).    We consider the incompressible flow in a pipe, where the governing equation is the incompressible Navier-Stokes equation, as following,

$$\frac{\partial v}{\partial t} + (v \cdot \nabla)v = -\nabla p + \mu \nabla^2 v, \quad \nabla \cdot v = 0$$

where $v$ is the velocity vector, $p$ is the pressure, and $\mu = 0.005$ is the viscosity. The parabolic velocity profile with maximum velocity $v = [1, 0]$ is imposed at the inlet. A free boundary condition is imposed at the outlet, and a no-slip boundary condition is imposed at the pipe surface. The pipe has a length of 10 and width of 1. The centerline of the pipe is parameterized by 4 piecewise cubic polynomials, which are determined by the vertical positions and slopes on 5 spatially uniform control nodes. The vertical position at these control nodes obeys $d \sim \mathcal{U}[-2, 2]$, and the slope at these control nodes obeys $d \sim \mathcal{U}[-1, 1]$.

We have access to 1000 training data and 200 test data, generated with an implicit finite element solver using about 4000 Taylor-Hood Q2-Q1 mixed elements. The mesh point locations ($129 \times 129$) and horizontal velocity on these mesh points are used as input and output data.

# B  Implementation Details

We implemented all experiments with PyTorch (Paszke et al., 2019). The code is available at https://anonymous.4open.science/r/coral-0348/. We estimate the computation time needed for development and the different experiments to approximately 400 days.

## B.1  CORAL

### B.1.1  Architecture Details

**SIREN initialization.** We use for SIREN the same initialization scheme as in Sitzmann et al. (2020b), i.e., sampling the weights of the first layer according to a uniform distribution $\mathcal{U}(-1/d, 1/d)$ and the next layers according to $\mathcal{U}(-\frac{1}{w_0}\sqrt{\frac{6}{d_{in}}}, \frac{1}{w_0}\sqrt{\frac{6}{d_{in}}})$. We use the default PyTorch initialization for the hypernetwork.

**Decode with shift-modulated SIREN.** Initially, we attempted to modulate both the scale and shift of the activation, following the approach described in Perez et al. (2018). However, we did not observe any performance improvement by employing both modulations simultaneously. Consequently, we decided to focus solely on shift modulations, as it led to a more stable training process and reduced the size of the modulation space by half. We provide an overview of the decoder with the shift-modulated SIREN in Figure 3.

**Encode with *auto-decoder*.** We provide a schematic view of the input encoder in Figure 4. The *auto-decoding* process starts from a code $z_a = 0$ and performs $K$ steps of gradient descent over this latent code to minimize the reconstruction loss.

**Process with MLP.** We use an MLP with skip connections and Swish activation functions. Its forward function writes $g_\psi(z) = \mathrm{Block}_k \circ ... \circ \mathrm{Block}_1(z)$, where Block is a two-layer MLP with skip connections:

$$\mathrm{Block}(z) = z + \sigma(\mathbf{W}_2 \cdot \sigma(\mathbf{W}_1 \cdot z + \mathbf{b}_1) + \mathbf{b}_2) \tag{15}$$

In Equation (15), $\sigma$ denotes the feature-wise Swish activation. We use the version with learnable parameter $\beta$; $\sigma(z) = z \cdot \mathrm{sigmoid}(\beta z)$.

### B.1.2  Training Details

The training is done in two steps. First, we train the modulated INRs to represent the data. We show the details with the pseudo-code in Algorithms 1 and 2. $\alpha$ is the inner-loop learning rate while $\lambda$ is the outer loop learning rate, which adjusts the weights of the INR and hypernetwork. Then, once the INRs have been fitted, we obtain the latent representations of the training data, and use these latent codes to train the forecast model $g_\psi$ (See Algorithm 3). We note $\lambda_\psi$ the learning rate of $g_\psi$.

**Z-score normalization.** As the data is encoded using only a few steps of gradients, the resulting standard deviation of the codes is very small, falling within the range of [1e-3, 5e-2]. However, these "raw" latent representations are not suitable as-is for further processing. To address this, we normalize the codes by subtracting the mean and dividing by the standard deviation, yielding the normalized code: $z_{\mathrm{norm}} = \frac{z - \mathrm{mean}}{\mathrm{std}}$. Depending on the task, we employ slightly different types of normalization:

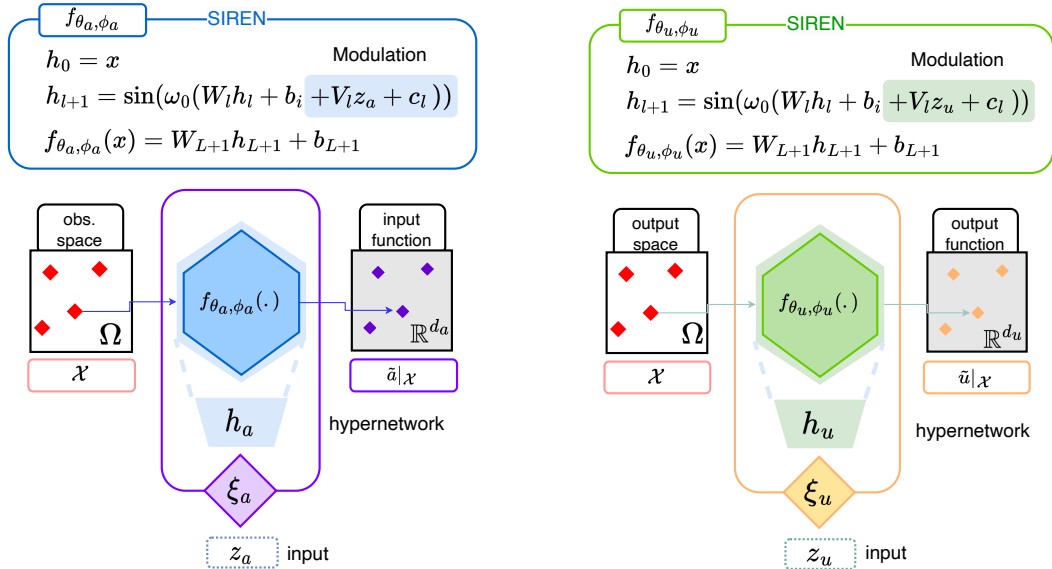

(a) The hypernetwork $h_a$ maps the input code $z_a$ to the modulations $\phi_a$. The modulations shift the activations at each layer of the SIREN.

(b) The hypernetwork $h_u$ maps the input code $z_u$ to the modulations $\phi_u$. The modulations shift the activations at each layer of the SIREN.

Figure 3: Architecture of the input and output decoders $\xi_a, \xi_u$. They can be queried on any coordinate $x \in \Omega$. We use the same notation for both, even though the parameters are different.

1. Initial value problem: • *Cylinder*: We normalize the inputs and outputs code with the same mean and standard deviation. We compute the statistics feature-wise, across the inputs and outputs. • *Airfoil*: We normalize the inputs and outputs code with their respective mean and standard deviation. The statistics are real values.

2. Dynamics modeling: We normalize the codes with the same mean and standard deviation. The statistics are computed feature-wise, over all training trajectories and all available timestamps (i.e. over $In\text{-}t$).

3. Geometry-aware inference: We normalize the input codes only, with feature-wise statistics.

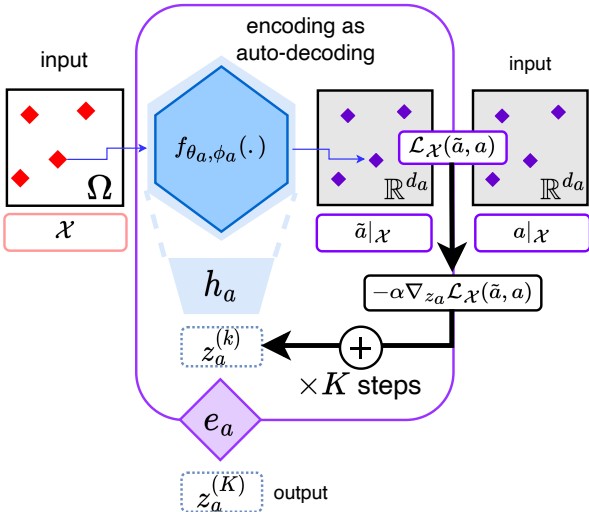

Figure 4: Starting from a code $z_a^{(0)} = 0$, the input encoder $e_a$ performs $K$ inner steps of gradient descent over $z_a$ to minimize the reconstruction loss $\mathcal{L}_\mathcal{X}(\tilde{a}, a)$ and outputs the resulting code $z_a^{(K)}$ of this optimization process. During training, we accumulate the gradients of this encoding phase and back-propagate through the $K$ inner-steps to update the parameters $\theta_a$ and $w_a$. At inference, we encode new inputs with the same number of steps $K$ and the same learning rate $\alpha$, unless stated otherwise. The output encoder works in the same way during training, and is not used at inference.

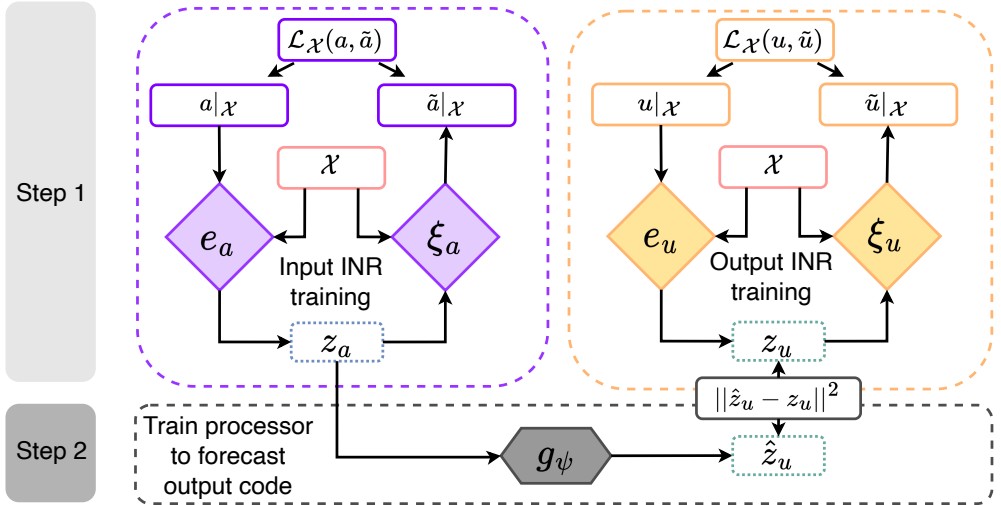

Figure 5: Proposed training for CORAL. (1) We first learn to represent the data with the input and output INRs. (2) Once the INRs are trained, we obtain the latent represenations and fix the pairs of input and output codes $(z_{a_i}, z_{u_i})$. We then train the processor to minimize the distance between the processed code $g_\psi(z_{a_i})$ and the output code $z_{u_i}$.

### B.1.3 Inference Details

We present the inference procedure in Algorithm 4. It is important to note that the input and output INRs, $f_{\theta_a}$ and $f_{\theta_u}$, respectively, accept the "raw" codes as inputs, whereas the processor expects a normalized latent code. Therefore, after the encoding steps, we normalize the input code. Additionally, we may need to denormalize the code immediately after the processing stage. It is worth mentioning that we maintain the same number of inner steps as used during training, which is 3 for all tasks.

**Algorithm 1:** Training of the input INR

**while** no convergence **do**
    Sample batch $\mathcal{B}$ of data $(a_i)_{i \in \mathcal{B}}$;
    Set codes to zero $z_{a_i} \leftarrow 0, \forall i \in \mathcal{B}$ ;
    **for** $i \in \mathcal{B}$ and step $\in \{1, ..., K_a\}$ **do**
        $z_{a_i} \leftarrow$
        $z_{a_i} - \alpha_a \nabla_{z_{a_i}} \mathcal{L}_{\mathcal{X}_i}(f_{\theta_a, h_a(z_{a_i})}, a_i)$ ;
        `// input encoding inner`
        `step`
    **end**
    `/* outer loop update       */`
    $\theta_a \leftarrow \theta_a -$
    $\lambda \frac{1}{|\mathcal{B}|} \sum_{i \in \mathcal{B}} \nabla_{\theta_a} \mathcal{L}_{\mathcal{X}_i}(f_{\theta_a, h_a(z_{a_i})}, a_i)$;
    $w_a \leftarrow w_a -$
    $\lambda \frac{1}{|\mathcal{B}|} \sum_{i \in \mathcal{B}} \nabla_{w_a} \mathcal{L}_{\mathcal{X}_i}(f_{\theta_a, h_a(z_{a_i})}, a_i)$
**end**

**Algorithm 2:** Training of the output INR

**while** no convergence **do**
    Sample batch $\mathcal{B}$ of data $(a_i, u_i)_{i \in \mathcal{B}}$;
    Set codes to zero $z_{u_i} \leftarrow 0, \forall i \in \mathcal{B}$ ;
    **for** $i \in \mathcal{B}$ and step $\in \{1, ..., K_u\}$ **do**
        $z_{u_i} \leftarrow$
        $z_{u_i} - \alpha_u \nabla_{z_{u_i}} \mathcal{L}_{\mathcal{X}_i}(f_{\theta_u, h_u(z_{u_i})}, u_i)$
        ; `// output encoding inner`
        `step`
    **end**
    `/* outer loop update       */`
    $\theta_u \leftarrow \theta_u -$
    $\lambda \frac{1}{|\mathcal{B}|} \sum_{i \in \mathcal{B}} \nabla_{\theta_u} \mathcal{L}_{\mathcal{X}_i}(f_{\theta_u, h_u(z_{u_i})}, u_i)$;
    $w_u \leftarrow w_u -$
    $\lambda \frac{1}{|\mathcal{B}|} \sum_{i \in \mathcal{B}} \nabla_{w_u} \mathcal{L}_{\mathcal{X}_i}(f_{\theta_u, h_u(z_{u_i})}, u_i)$
**end**

---

**Algorithm 3:** Training of the processor

**while** no convergence **do**
    Sample batch $\mathcal{B}$ of codes $(z_{a_i}, z_{u_i})_{i \in \mathcal{B}}$;
    `/* processor update                                        */`
    $\psi \leftarrow \psi - \lambda_\psi \frac{1}{|\mathcal{B}|} \sum_{i \in \mathcal{B}} \nabla_\psi \mathcal{L}(g_\psi(z_{a_i}), z_{u_i})$ ;
**end**

---

**Algorithm 4:** CORAL Inference, given a function $a$

Set code to zero $z_a \leftarrow 0$ ;
**for** step $\in \{1, ..., K_a\}$ **do**
    $z_a \leftarrow z_a - \alpha_a \nabla_{z_a} \mathcal{L}_{\mathcal{X}}(f_{\theta_a, h_a(z_a)}, a)$ ;        `// input encoding inner step`
**end**
$\hat{z}_u = g_\psi(z_a)$ ;                               `// process latent code`
$\hat{u} = f_{\theta_u, h_u(\tilde{z}_u)}$ ;                           `// decode output function`

### B.1.4 Choice of Hyperparameters

We recall that $d_z$ denotes the size of the code, $w_0$ is a hyperparameter that controls the frequency bandwith of the SIREN network, $\lambda$ is the outer-loop learning rate (on $f_{\theta,\phi}$ and $h_w$), $\alpha$ is the inner-loop learning rate, $K$ is the number of inner steps used during training and encoding steps at test time, $\lambda_\psi$ is the learning rate of the MLP or NODE. In some experiments we learn the inner-loop learning rate $\alpha$, as in Li et al. (2017). In such case, the meta-$\alpha$ learning rate is an additional parameter that controls how fast we move $\alpha$ from its initial value during training. When not mentioned we simply report $\alpha$ in the tables below, and otherwise we report the initial learning rate and this meta-learning-rate.

We use the Adam optimizer during both steps of the training. For the training of the Inference / Dynamics model, we use a learning rate scheduler which reduces the learning rate when the loss has stopped improving. The threshold is set to 0.01 in the default relative threshold model in PyTorch, with a patience of 250 epochs w.r.t. the train loss. The minimum learning rate is 1e-5.

**Initial Value Problem** We provide the list of hyperparameters used for the experiments on *Cylinder* and *Airfoil* in Table 4.

**Dynamics Modeling** Table 5 summarizes the hyperparameters used in our experiments for dynamics modeling on datasets *Navier-Stokes* and *Shallow-Water* (Table 2).

Furthermore, to facilitate the training of the dynamics within the NODE, we employ Scheduled Sampling, following the approach described in Bengio et al. (2015). At each timestep, there is a

Table 4: **CORAL hyper-parameters for IVP/ Geometry-aware inference**

| | Hyper-parameter | *Cylinder* | *Airfoil* | *NACA-Euler* | *Elasticity* | *Pipe* |
|---|---|---|---|---|---|---|
| $f_{\theta_a,\phi_a}$ / $f_{\theta_u,\phi_u}$ | $d_z$ | 128 | 128 | 128 | 128 | 128 |
| | depth | 4 | 5 | 4 | 4 | 5 |
| | width | 256 | 256 | 256 | 256 | 128 |
| | $\omega_0$ | 30 | 30 / 50 | 5 / 15 | 10 / 15 | 5 / 10 |
| SIREN Optimization | batch size | 32 | 16 | 32 | 64 | 16 |
| | epochs | 2000 | 1500 | 5000 | 5000 | 5000 |
| | $\lambda$ | 5e-6 | 5e-6 | 1e-4 | 1e-4 | 5e-5 |
| | $\alpha$ | 1e-2 | 1e-2 | 1e-2 | 1e-2 | 1e-2 |
| | meta-$\alpha$ learning rate | 0 | 5e-6 | 1e-4 | 1e-4 | 5e-5 |
| | $K_a$ / $K_u$ | 3 | 3 | 3 | 3 | 3 |
| $g_\psi$ | depth | 3 | 3 | 3 | 3 | 3 |
| | width | 64 | 64 | 64 | 64 | 128 |
| | activation | Swish | Swish | Swish | Swish | Swish |
| Inference Optimization | batch size | 32 | 16 | 64 | 64 | 64 |
| | epochs | 100 | 100 | 10000 | 10000 | 10000 |
| | $\lambda_\psi$ | 1e-3 | 1e-3 | 1e-3 | 1e-3 | 1e-3 |
| | Scheduler decay | 0 | 0 | 0.9 | 0.9 | 0.9 |

Table 5: **CORAL hyper-parameters for dynamics modeling**

| | Hyper-parameter | *Navier-Stokes* | *Shallow-Water* |
|---|---|---|---|
| INR | $d_z$ | 128 | 256 |
| | depth | 4 | 6 |
| | width | 128 | 256 |
| | $\omega_0$ | 10 | 10 |
| INR Optimization | batch size | 64 | 16 |
| | epochs | 10, 000 | 10, 000 |
| | $\lambda$ | 5e-6 | 5e-6 |
| | $\alpha$ | 1e-2 | 1e-2 |
| | $K$ | 3 | 3 |
| NODE | depth | 3 | 3 |
| | width | 512 | 512 |
| | activation | Swish | Swish |
| | solver | RK4 | RK4 |
| Dynamics Optimization | batch size | 32 | 16 |
| | epochs | 10, 000 | 10, 000 |
| | $\lambda_\psi$ | 1e-3 | 1e-3 |
| | Scheduler decay | 0.75 | 0.75 |

probability of $\epsilon\%$ for the integration of the dynamics through the ODE solver to be restarted using the training snapshots. This probability gradually decreases during the training process. Initially, we set $\epsilon_{\text{init}} = 0.99$, and every 10 epochs, we multiply it by 0.99. Consequently, by the end of the training procedure, the entire trajectory is computed with the initial condition.

**Geometry-aware inference**  We provide the list of hyperparameters used for the experiments on *NACA-Euler*, *Elasticity*, and *Pipe* in Table 4.

## B.2  Baseline Implementation

We detail in this section the architecture and hyperparameters used for the training of the baselines presented in Section 4.

**Initial Value Problem**  We use the following baselines for the Initial Value Problem task.

- **NodeMLP**. We use a ReLU-MLP with 3 layers and 512 neurons. We train it for 10000 epochs. We use a learning rate of 1e-3 and a batch size of 64.

- **GraphSAGE**. We use the implementation from torch-geometric (Fey & Lenssen, 2019), with 6 layers of 64 neurons. We use ReLU activation. We train the model for 400 epochs for *Airfoil* and 4,000 epochs for *Cylinder*. We build the graph using the 16 closest nodes. We use a learning rate of 1e-3 and a batch size of 64.

- **MP-PDE**: We implement MP-PDE as a 1-step solver, where the time-bundling and pushforward trick do not apply. We use 6 message-passing blocks and 64 hidden features. We build the graph with the 16 closest nodes. We use a learning rate of 1e-3 and a batch size of 16. We train for 500 epochs on *Airfoil* and 1000 epochs on *Cylinder*.

**Dynamics Modeling**  All our baselines are implemented in an auto-regressive (AR) manner to perform forecasting.

- **DeepONet**: We use a DeepONet in which both Branch Net and Trunk Net are 4-layers MLP with 100 neurons. The model is trained for 10,000 epochs with a learning rate of 1e-5. To complete the upsampling studies, we used a modified DeepONet forward which computes as follows: (1) Firstly, we compute an AR pass on the training grid to obtain a prediction of the complete trajectory with the model on the training grid. (2) We use these prediction as input of the branch net for a second pass on the up-sampling grid to obtain the final prediction on the new grid.

- **FNO**: FNO is trained for 2,000 epochs with a learning rate of 1e-3. We used 12 modes and a width of 32 and 4 Fourier layers. We also use a step scheduler every 100 epochs with a decay of 0.5.

- **MP-PDE**: We implement MP-PDE with a time window of 1 so that is becomes AR. The MP-PDE solver is composed of a 6 message-passing blocks with 128 hidden features. To build the graphs, we limit the number of neighbors to 8. The optimization was performed on 10,000 epochs with a learning rate of 1e-3 and a step scheduler every 2000 epochs until 10000. We decay the learning rate of 0.4 with weight decay 1e-8.

- **DINo**: DINo uses MFN model with respectively width and depth of 64 and 3 for Navier-Stokes (NS), and 256 and 6 for Shallow-Water (SW). The encoder proceeds to 300 (NS) or 500 (SW) steps to optimize the codes whose size is set to 100 (NS) or 200 (SW). The dynamic is solved with a NODE that uses 4-layers MLP and a hidden dimension of 512 (NS) or 800 (SW). This model is trained for 10,000 epochs with a learning rate of 5e-3. We use the same scheduled sampling as for the CORAL training (see appendix B.1.4).

**Geometry-aware inference**  Except on *Pipe*, the numbers for **FactorizedFNO** are taken from Tran et al. (2023). In the latter we take the 12-layer version which has a comparable model size. We train the 12-layer FactorizedFNO on *Pipe* with AdamW for 200 epochs with modes (32, 16), a width of 64, a learning rate of 1e-3 and a weight decay of 1e-4. We implemented the baselines **GeoFNO**, **FNO**, **UNet** according to the code provided in Li et al. (2022a)

## C    Supplementary Results for Dynamics Modeling

### C.1    Robustness to Resolution Changes

We present in Tables 6 and 7 the up-sampling capabilities of CORAL and relevant baselines both *In-t* and *Out-t*, respectively for *Navier-Stokes* and *Shallow-Water*.

Table 6: **Up-sampling capabilities** - Test results on *Navier-Stokes* dataset. Metrics in MSE.

| $\mathcal{X}_{tr} \downarrow$ | dataset $\rightarrow$ $\mathcal{X}_{tr} \rightarrow$ $\mathcal{X}_{te} \rightarrow$ | | | | | | | | |
|---|---|---|---|---|---|---|---|---|---|
| | | \multicolumn Navier-Stokes $64 \times 64$ | | | | | | | |
| | | $\mathcal{X}_{tr}$ | | $64 \times 64$ | | $128 \times 128$ | | $256 \times 256$ | |
| | | *In-t* | *Out-t* | *In-t* | *Out-t* | *In-t* | *Out-t* | *In-t* | *Out-t* |
| | DeepONet | 1.47e-2 | 7.90e-2 | 1.47e-2 | 7.90e-2 | 1.82e-1 | 7.90e-2 | 1.82e-2 | 7.90e-2 |
| $\pi_{tr} = 100\%$ | FNO | 7.97e-3 | 1.77e-2 | 7.97e-3 | 1.77e-2 | 8.04e-3 | 1.80e-2 | 1.81e-2 | 7.90e-2 |
| regular grid | MP-PDE | 5.98e-4 | 2.80e-3 | 5.98e-4 | 2.80e-3 | 2.36e-2 | 4.61e-2 | 4.26e-2 | 9.77e-2 |
| | DINo | 1.25e-3 | 1.13e-2 | 1.25e-3 | 1.13e-2 | 1.25e-3 | 1.13e-2 | 1.26e-3 | 1.13e-2 |
| | CORAL | **2.02e-4** | **1.07e-3** | **2.02e-4** | **1.07e-3** | **2.08e-4** | **1.06e-3** | **2.19e-4** | **1.07e-3** |
| | DeepONet | 8.35e-1 | 7.74e-1 | 8.28e-1 | 7.74e-1 | 8.32e-1 | 7.74e-1 | 8.28e-1 | 7.73e-1 |
| $\pi_{tr} = 20\%$ | MP-PDE | 2.36e-2 | 1.11e-1 | 7.42e-2 | 2.13e-1 | 1.18e-1 | 2.95e-1 | 1.37e-1 | 3.39e-1 |
| irregular grid | DINo | **1.30e-3** | 9.58e-3 | **1.30e-3** | 9.59e-3 | **1.31e-3** | 9.63e-3 | **1.32-3** | 9.65e-3 |
| | CORAL | 1.73e-3 | **5.61e-3** | 1.55e-3 | **4.34e-3** | 1.61e-3 | **4.38e-3** | 1.65e-3 | **4.41e-3** |
| | DeepONet | 7.12e-1 | 7.16e-1 | 7.22e-1 | 7.26e-1 | 7.24e-1 | 7.28e-1 | 7.26e-1 | 7.30e-1 |
| $\pi_{tr} = 5\%$ | MP-PDE | 1.25e-1 | 2.92e-1 | 4.83e-1 | 1.08 | 6.11e-1 | 1.07 | 6.49e-1 | 1.08 |
| irregular grid | DINo | 8.21e-2 | 1.03e-1 | 7.73e-2 | 7.49e-2 | 7.87e-2 | 7.63e-2 | 7.96e-2 | 7.73e-2 |
| | CORAL | **1.56e-2** | **3.65e-2** | **4.19e-3** | **1.12e-2** | **4.30e-3** | **1.14e-2** | **4.37e-3** | **1.14e-2** |

Table 7: **Up-sampling capabilities** - Test results on *Shallow-Water* dataset. Metrics in MSE.

| $\mathcal{X}_{tr} \downarrow$ | dataset $\rightarrow$ $\mathcal{X}_{tr} \rightarrow$ $\mathcal{X}_{te} \rightarrow$ | | | | | | | | |
|---|---|---|---|---|---|---|---|---|---|
| | | \multicolumn Shallow-Water $64 \times 128$ | | | | | | | |
| | | $\mathcal{X}_{tr}$ | | $32 \times 64$ | | $64 \times 128$ | | $128 \times 256$ | |
| | | *In-t* | *Out-t* | *In-t* | *Out-t* | *In-t* | *Out-t* | *In-t* | *Out-t* |
| | DeepONet | 7.07e-3 | 9.02e-3 | 1.18e-2 | 1.66e-2 | 7.07e-3 | 9.02e-3 | 1.18e-2 | 1.66e-2 |
| $\pi_{tr} = 100\%$ | FNO | 6.75e-5 | **1.49e-4** | 7.54e-5 | **1.78e-4** | 6.75e-5 | **1.49e-4** | 6.91e-5 | **1.52e-4** |
| regular grid | MP-PDE | 2.66e-5 | 4.35e-4 | 4.80e-2 | 1.42e-2 | 2.66e-5 | 4.35e-4 | 4.73e-3 | 1.73e-3 |
| | DINo | 4.12e-5 | 2.91e-3 | 5.77e-5 | 2.55e-3 | 4.12e-5 | 2.91e-3 | 6.04e-5 | 2.58e-3 |
| | CORAL | **3.52e-6** | 4.99e-4 | **1.86e-5** | 5.32e-4 | **3.52e-6** | 4.99e-4 | **4.96e-6** | 4.99e-4 |
| | DeepONet | 1.08e-2 | 1.10e-2 | 2.49e-2 | 3.25e-2 | 2.49e-2 | 3.25e-2 | 2.49e-2 | 3.22e-2 |
| irregular grid | MP-PDE | 4.54e-3 | 1.48e-2 | 4.08e-3 | 1.30e-2 | 5.46e-3 | 1.74e-2 | 4.98e-3 | 1.43e-2 |
| $\pi_{tr} = 20\%$ | DINo | 2.32e-3 | 5.18e-3 | 2.22e-3 | 4.80e-3 | 2.16e-3 | 4.64e-3 | 2.16e-3 | 4.64e-3 |
| | CORAL | **1.36e-3** | **2.17e-3** | **1.24e-3** | **1.95e-3** | **1.21e-3** | **1.95e-3** | **1.21e-3** | **1.95e-3** |
| | DeepONet | 1.02e-2 | **1.01e-2** | 1.57e-2 | 1.93e-2 | 1.57e-2 | 1.93e-2 | 1.57e-2 | 1.93e-2 |
| irregular grid | MP-PDE | **5.36e-3** | 1.81e-2 | **5.53e-3** | 1.80e-2 | **4.33e-3** | 1.32e-2 | **5.48e-3** | 1.74e-2 |
| $\pi_{tr} = 5\%$ | DINo | 1.25e-2 | 1.51e-2 | 1.39e-2 | 1.54e-2 | 1.39e-2 | 1.54e-2 | 1.39e-2 | 1.54e-2 |
| | CORAL | 8.40e-3 | 1.25e-2 | 9.27e-3 | **1.15e-2** | 9.26e-3 | **1.16e-2** | 9.26e-3 | **1.16e-2** |

These tables show that CORAL remains competitive and robust on up-sampled inputs. Other baselines can also predict on denser grids, except for MP-PDE, which over-fitted the training grid.

### C.2    Learning a Dynamics on Different Grids

To extend our work, we propose to study how robust is CORAL to changes in grids. In our classical setting, we keep the same grid for all trajectories in the training set and evaluate it on a new grid for the test set. Instead, here, both in train and test sets, each trajectory $i$ has its own grid $\mathcal{X}_i$. Thus, we evaluate CORAL's capability to generalize to grids. We present the results in Table 8. Overall, coordinate-based methods generalize better over grids compared to operator based and discrete methods like DeepONet and MP-PDE which show better or equivalent performance when trained only on one grid. CORAL's performance is increased when trained on different grids; one possible reason is that CORAL overfits the training grid used for all trajectories in our classical setting.

Table 8: **Learning dynamics on different grids** - Test results in the extrapolation setting. Metrics in MSE.

| $\mathcal{X}_{tr} \downarrow \mathcal{X}_{te}$ | dataset → | *Navier-Stokes* | | *Shallow-Water* | |
|---|---|---|---|---|---|
| | | *In-t* | *Out-t* | *In-t* | *Out-t* |
| $\pi = 20\%$ irregular grid | DeepONet | 5.22E−1 | 5.00E−1 | 1.11E−2 | 1.12E−2 |
| | MP-PDE | 6.11E−1 | 6.10E−1 | 6.80E−3 | 1.87E−2 |
| | DINo | 1.30E−3 | 1.01E−2 | 4.12E−4 | 3.05E−3 |
| | CORAL | **3.21E−4** | **3.03E−3** | **1.15E−4** | **7.75E−4** |
| $\pi = 5\%$ irregular grid | DeepONet | 4.11E−1 | 4.38E−1 | 1.11E−2 | 1.12E−2 |
| | MP-PDE | 8.15E−1 | 1.10E−1 | 1.22E−2 | 4.29E−2 |
| | DINo | 1.26E−3 | 1.04E−2 | 3.89E−3 | 7.41E−3 |
| | CORAL | **9.82E−4** | **9.71E−3** | **2.22e-3** | **4.89e-3** |

Table 9: **Training time comparison** - Expressed in days (d) or hours (h) on several datasets.

| Model | *Cylinder* | *Navier-Stokes* | *Shallow-Water* | *Elasticity* | *NACA* |
|---|---|---|---|---|---|
| CORAL (INR) | 6h | 1d | 5d | 4h | 2d |
| CORAL (Process) | 1h | 4h | 1h | 1h | 1h |
| NodeMLP | 0.5h | - | - | - | - |
| GraphSAGE | 1d | - | - | - | - |
| MP-PDE | 7h | 19h | 21h | - | - |
| DINo | - | 8h | 2d | - | - |
| DeepONet | - | 6h | 5h | - | - |
| FNO | - | 8h | 6h | 0.5h | 0.5h |
| UNet | - | - | - | 0.5h | 0.5h |
| Geo-FNO | - | - | - | 1.0h | 1.0h |
| Factorized-FNO | - | - | - | 1.0h | 1.0h |

## C.3 Training Time

In Table 9, we present the training time for CORAL and different baselines for comparison. Since, the training of CORAL is separated in 2 steps, we show in line "INR" the training time for INR fitting and in line "Process" the second step to train the forecast model. We see that the longest part of the training procedure in CORAL is the fitting of the INR. MP-PDE is the slowest baseline to train, due to the KNN graph creation. DeepONet and FNO are the fastest baselines to train because they only need a forward pass.

## C.4 Inference Time

In this section, we evaluate the inference time of CORAL and other baselines w.r.t. the input grid size. We study the impact of the training grid size (different models trained with 5%, 20% and 100% of the grid) (Figure 6a) and the time needed for a model trained (5%) on a given grid to make computation on finer grid size resolution (evaluation grid size) (Figure 6b).

On the graphs presented in Figure 6, we observe that except for the operator baselines, CORAL is also competitive in terms of inference time. MP-PDE inference time increases strongly when inference grid gets denser. The DINo model, which is the only to propose the same properties as CORAL, is much slower when both inference and training grid size evolve. This difference is mainly explained by the number of steps needed to optimize DINo codes. Indeed, at test time DINO's adaptation requires 100x more optimization steps. MPPDE's computation is slow due to the KNN graph creation and slower message passing. DeepONet and FNO are faster due to forward computation only. CORAL's encoding/decoding is relatively resolution-insensitive and performed in parallel across all sensors. Process operates on a fixed dimension independent of the resolution. FNO is fast due to FFT but

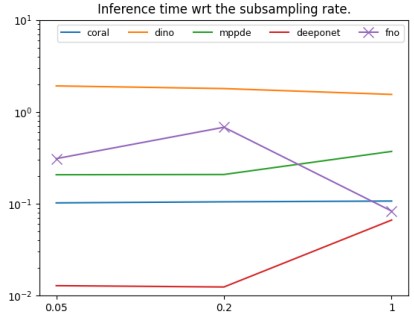 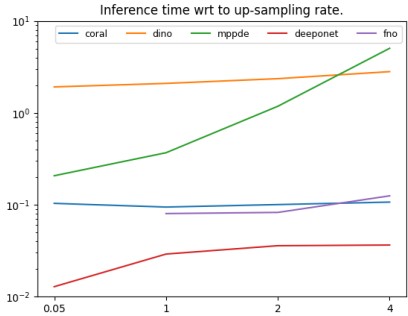

(a) Average inference time (in seconds) of the implemented baselines to unroll a trajectory until $T = 40$ on *Navier-Stokes* . For irregular grids, FNO is performed following linear interpolation.

(b) Inference time w.r.t. evaluation grid size (same models).

Figure 6

cannot be used on irregular grids. For these experiments, we used linear interpolation which slowed the inference time.

## C.5  Propagation of Errors Through Time

In Figures 7a to 7c, we show the evolution of errors as the extrapolation horizon evolves. First, we observe that all baselines propagate error through time, since the trajectories are computed using an auto-regressive approach. Except for the $100\%$, DeepONet had difficulties to handle the dynamic. It has on all settings the highest error. Then, we observe that for MP-PDE and FNO, the error increases quickly at the beginning of the trajectories. This means that these two models are rapidly propagating error. Finally, both DINo and CORAL have slower increase of the error during *In-t* and *Out-t* periods. However, we clearly see on the graphs that DINo has more difficulties than CORAL to make predictions out-range. Indeed, while CORAL's error augmentation remains constant as long as the time evolves, DINo has a clear increase.

## C.6  Benchmarking INRs for CORAL

We provide some additional experiments for dynamics modeling with CORAL, but with diffrents INRs: MFN (Fathony et al., 2021), BACON (Lindell et al., 2022) and FourierFeatures (Tancik et al., 2020). Experiments have been done on Navier-Stokes on irregular grids sampled from grids of size $128 \times 128$. All training trajectories share the same grid and are evaluated on a new grid for test trajectories. Results are reported in Table 10. Note that we used the same learning hyper-parameters for the baselines than those used for SIREN in CORAL. SIREN seems to produce the best codes for dynamics modeling, both for in-range and out-range prediction.

Table 10: **CORAL results with different INRs.** - Test results in the extrapolation setting on *Navier-Stokes* dataset. Metrics in MSE.

| $\mathcal{X}_{tr} \downarrow \mathcal{X}_{te}$ | INR | *In-t* | *Out-t* |
|---|---|---|---|
| $\pi = 20\%$ irregular grid | SIREN | **5.76e-4** | **2.57e-3** |
| | MFN | 2.21e-3 | 5.17e-3 |
| | BACON | 2.90e-2 | 3.32e-2 |
| | FourierFeatures | 1.70e-3 | 5.67e-3 |
| $\pi = 5\%$ irregular grid | SIREN | **1.81e-3** | **4.15e-3** |
| | MFN | 9.97e-1 | 9.58e-1 |
| | BACON | 1.06 | 8.06e-1 |
| | FourierFeatures | 3.60e-1 | 3.62e-1 |

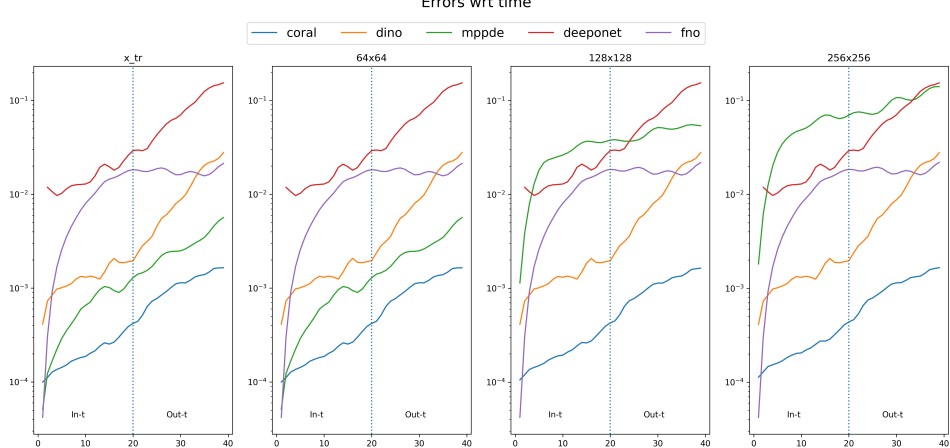

(a) Evolution of errors over time and across test samples for a model trained on 100% of the grid.

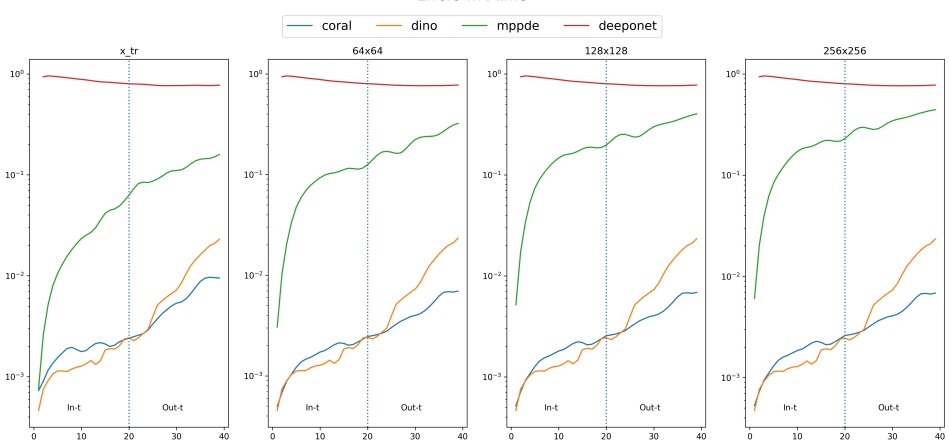

(b) Evolution of errors over time and across test samples for a model trained on 20% of the grid.

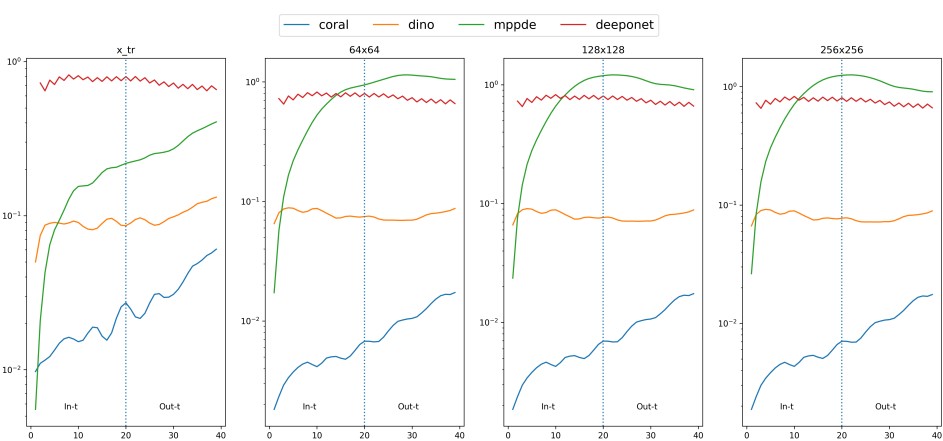

(c) Evolution of errors over time and across test samples for a model trained on 5% of the grid.

Figure 7: Errors along a given trajectory.

## C.7 Impact of 2nd order meta-learning

We provide in Figure 8 the evolution of the reconstruction error through the training epochs for *Navier-Stokes* with first-order and second-order meta-learning. The first order method is able to correctly train until it falls into an unstable regime for the common parameters. The second order method is much more stable and achieves a 10x reduction in MSE.

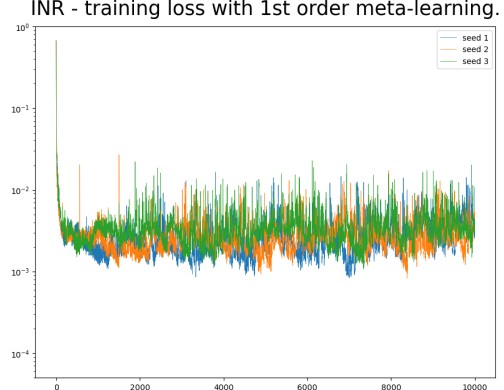 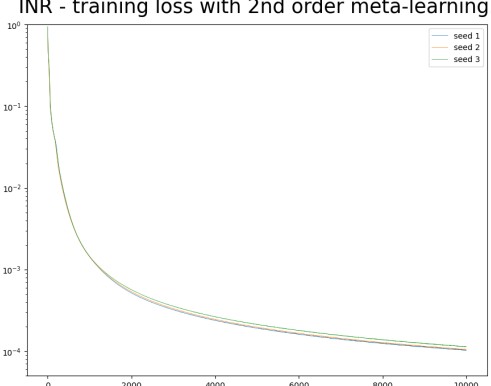

Figure 8: **Training of the modulated INR** - Comparison on *Navier-Stokes* over three independent runs of first order and second order meta-learning. We use the same number of inner-steps.

## C.8 Key hyper parameter analysis

Table 11 presents a hyperparameter study on reconstruction and forecasting tasks for *Navier-Stokes* dataset. Three hyperparameters—initial weight $w_0$, latent dimension $d_z$, and width—are varied to observe their impact. We can notice that:

- $w_0 = 30$ slightly improves the reconstruction on the test set.
- $d_z = 64$ yields a better forecasting In-t performance.
- width $= 256$ significantly improves the model's performance across nearly all metrics.

Table 11: **Hyper parameter study** - Reconstruction and forecasting results on *Navier-Stokes* dataset. Metrics in MSE. Reconstruction are reported on the Train (*In-t*) and on the Test (*In-t + Out-t*).

| Param ↓ | Value ↓ | Reconstruction | | Forecasting | |
|---|---|---|---|---|---|
| | | *Train* | *Test* | *In-t* | *Out-t* |
| $w_0$ | 20 | 3,62e-5 | 6,86e-5 | 2,78e-4 | 1,88e-3 |
| | 30 | 3,66e-5 | **5,85e-5** | 4,03e-4 | 2,28e-3 |
| $d_z$ | 64 | 3,94e-5 | 1,11e-4 | **1,22e-4** | 1,42e-3 |
| | 256 | 2,73e-5 | 8,03e-5 | 1,63e-4 | 2,12e-3 |
| width | 64 | 1,50e-4 | 2,87e-4 | 2,84e-4 | 2,39e-3 |
| | 256 | **1,60e-5** | 6,41e-5 | 1,23e-4 | 2,04e-3 |
| CORAL baseline | - | 1.05e-4 | 1.21e-4 | 1.86e-4 | **1.02e-3** |

## D Supplementary results for geometry-aware inference

### D.1 Inverse Design for NACA-airfoil

Once trained on *NACA-Euler*, CORAL can be used for the inverse design of a NACA airfoil. We consider an airfoil's shape parameterized by seven spline nodes and wish to minimize drag and maximize lift. We optimize the design parameters in an end-to-end manner. The spline nodes create

the input mesh, which CORAL maps to the output velocity field. This velocity field is integrated to compute the drag and the lift, and the loss objective is the squared drag over lift ratio. As can be seen in Figure 9, iterative optimization results in an asymmetric airfoil shape, enhancing progressively the lift coefficient in line with physical expectations. At the end of the optimization we reach a drag value of 0.042 and lift value of 0.322.

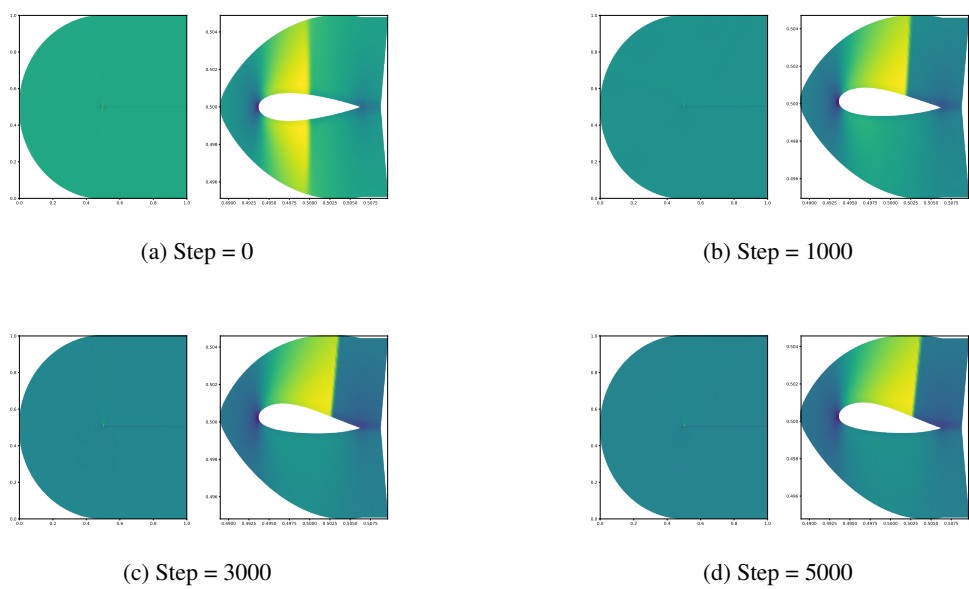

(a) Step = 0

(b) Step = 1000

(c) Step = 3000

(d) Step = 5000

Figure 9: Design optimization of a NACA-Airfoil.

# E   Qualitative results

In this section, we show different visualization of the predictions made by CORAL on the three considered tasks in this paper.

## E.1   Initial Value Problem

We provide in Figure 10 and Figure 11 visualizations of the inferred values of CORAL on *Cylinder* and *Airfoil*.

## E.2   Dynamics modeling

We provide in Figure 13 and Figure 12 visualization of the predicted trajectories of CORAL on *Navier-Stokes* and *Shallow-Water*.

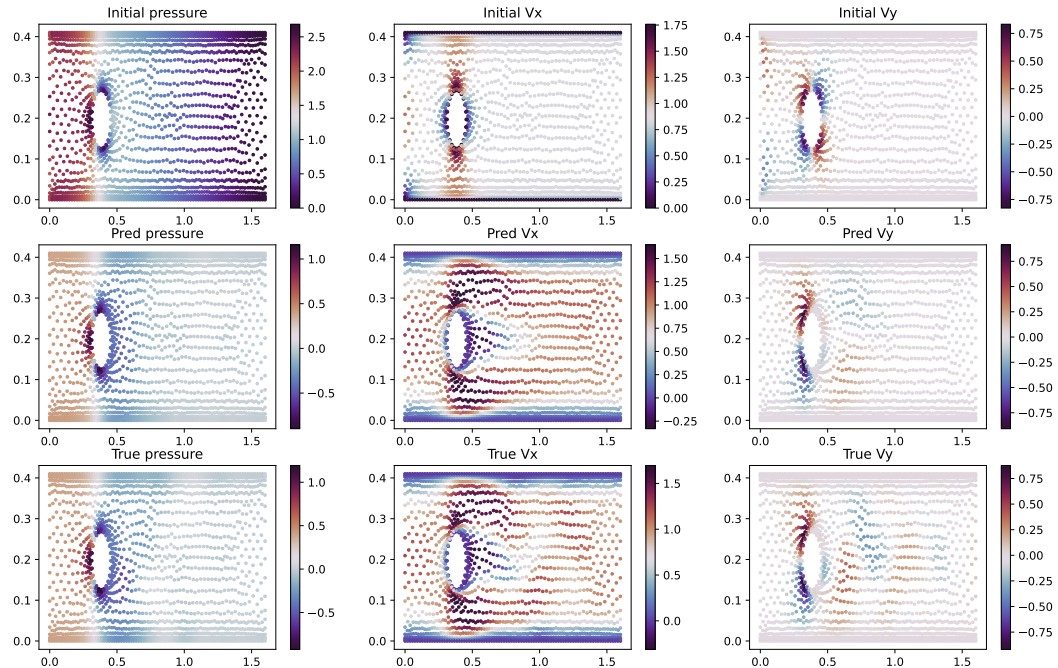

Figure 10: CORAL prediction on *Cylinder*

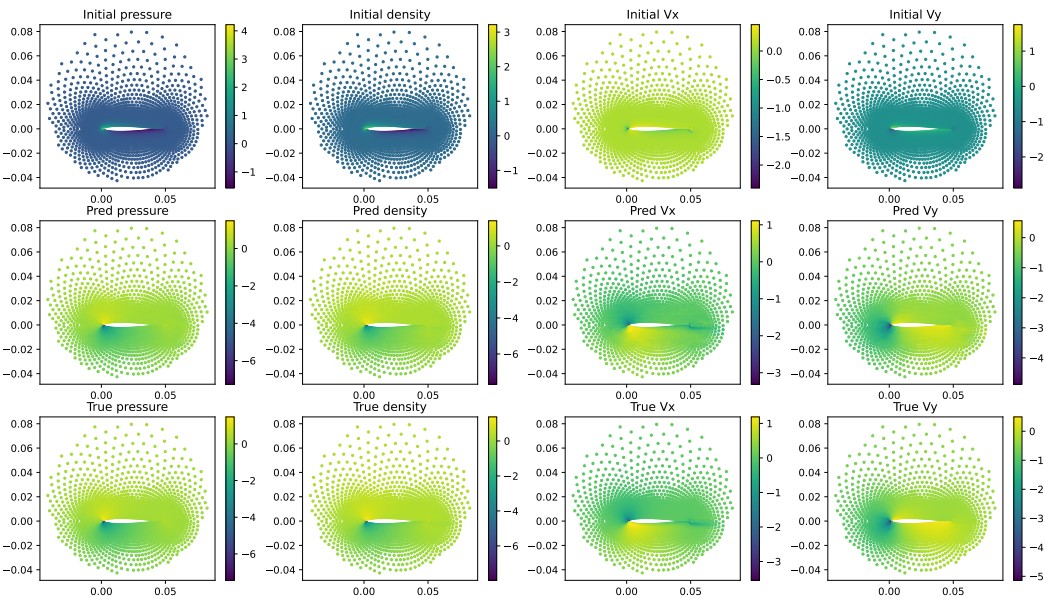

Figure 11: CORAL prediction on *Airfoil*

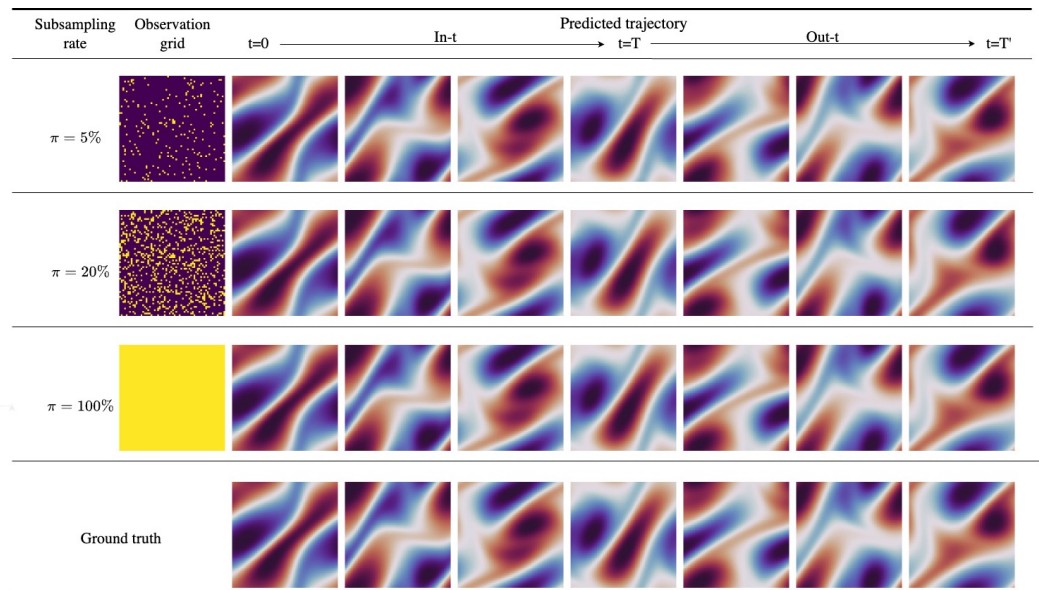

Figure 12: Prediction MSE per frame for CORAL on *Navier-Stokes* with its corresponding training grid $\mathcal{X}$. Each row corresponds to a different sampling rate and the last row is the ground truth. The predicted trajectory is predicted from $t = 0$ to $t = T'$. In our setting, $T = 19$ and $T' = 39$.

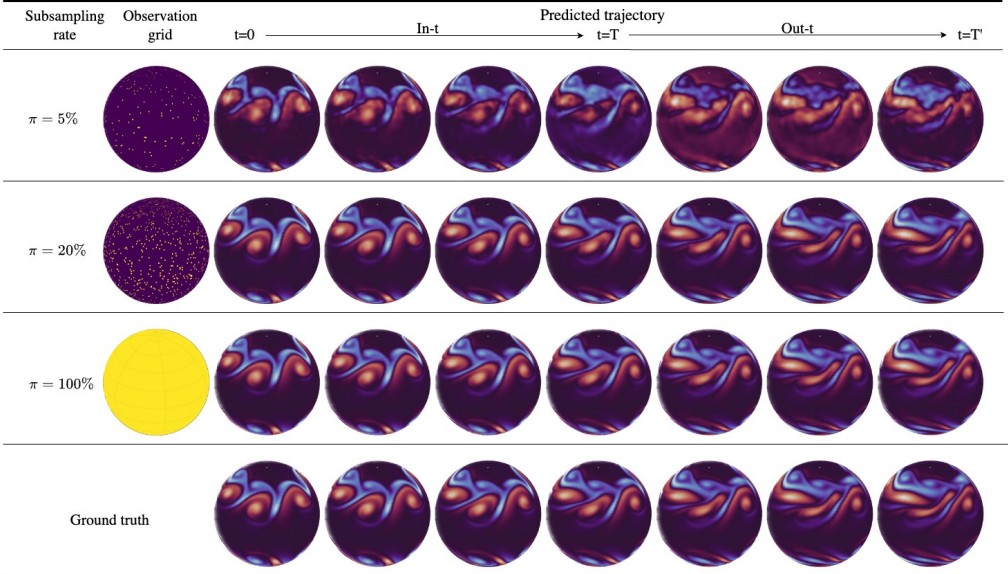

Figure 13: Prediction MSE per frame for CORAL on *Shallow-Water* with its corresponding training grid $\mathcal{X}$. Each row corresponds to a different sampling rate and the last row is the ground truth. The predicted trajectory is predicted from $t = 0$ to $t = T'$. In our setting, $T = 19$ and $T' = 39$.

## E.3 Geometry-aware inference

We provide in Figure 14, Figure 15, Figure 16 visualization of the predicted values of CORAL on *NACA-Euler*, *Pipe* and *Elasticity*.

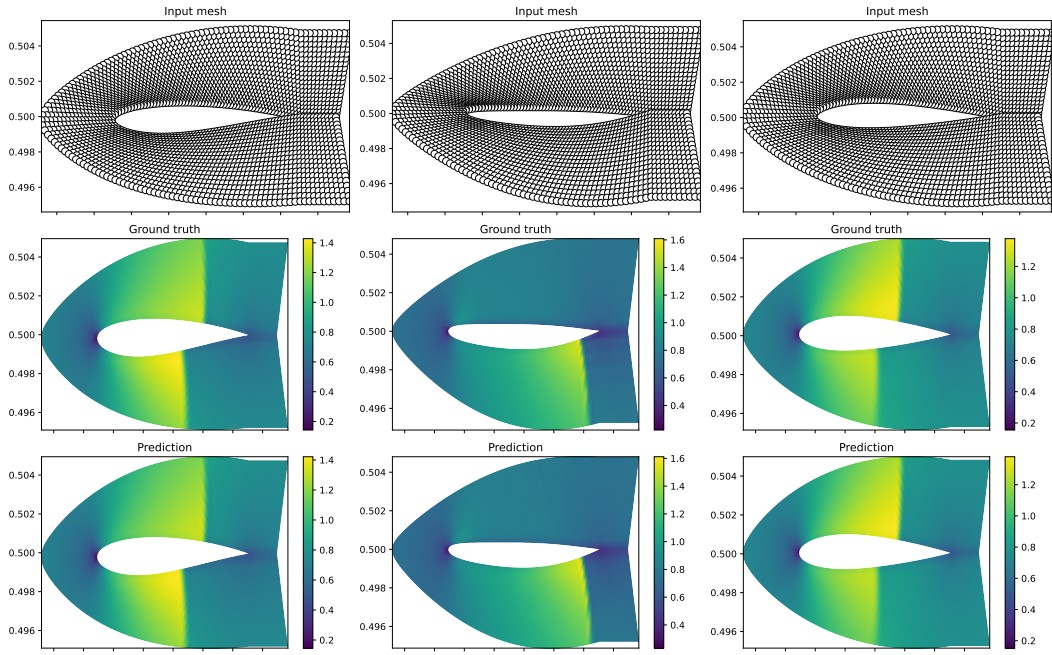

Figure 14: CORAL predictions on *NACA-Euler*

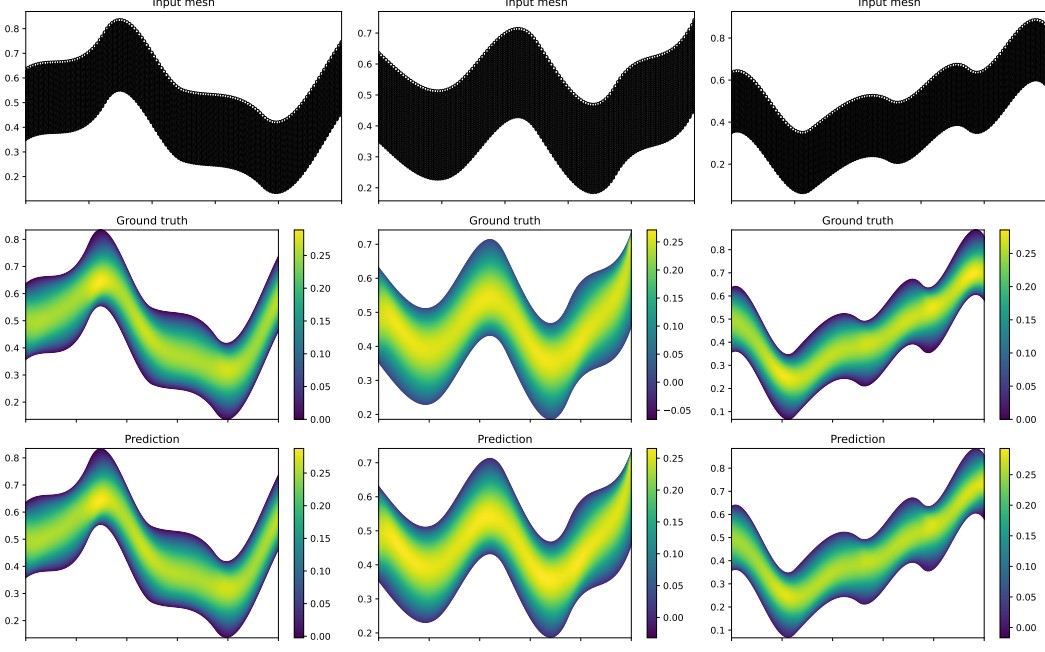

Figure 15: CORAL predictions on *Pipe*

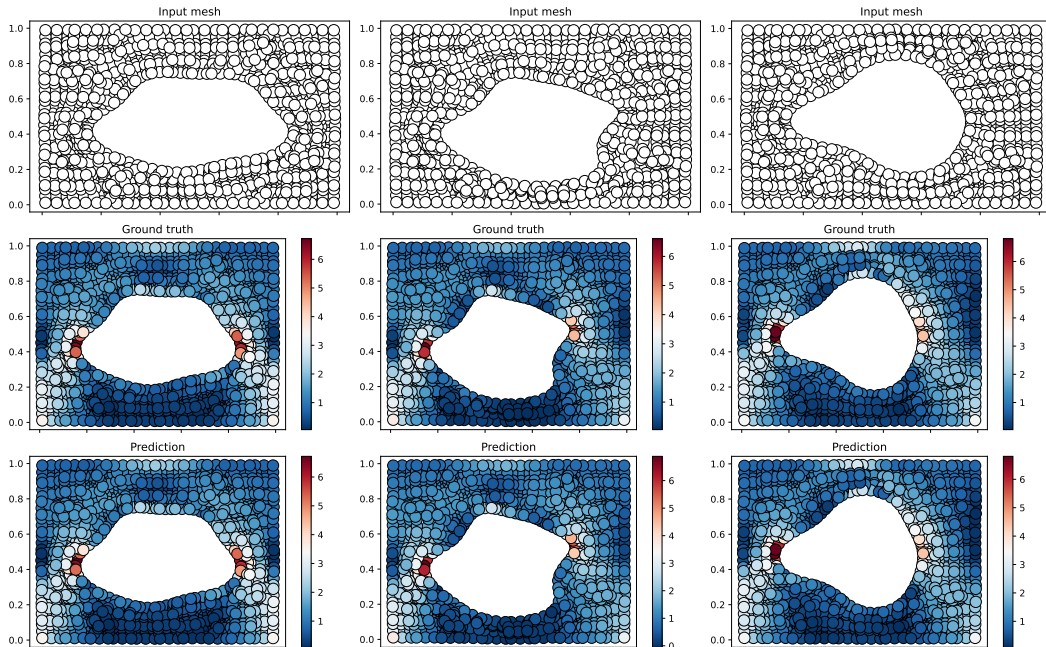

Figure 16: CORAL predictions on *Elasticity*

