# OpenReview forum: "Operator Learning with Neural Fields: Tackling PDEs on General Geometries"
_NeurIPS.cc/2023/Conference — NeurIPS 2023 poster_

### Official Review · Reviewer_K5kw · 2023-06-15

**Soundness:** 4 excellent
**Presentation:** 4 excellent
**Contribution:** 4 excellent
**Rating:** 6
**Confidence:** 5

**Summary:**

This paper creatively employs Implicit Neural Representations (INR) for operator learning on irregular domains. This novel method benefits from INR's capability to adaptively handle irregular grid distributions or irregular geometric areas, making learning the mapping from input to output in the INR's latent space quite natural. The primary experiments of the paper are on initial value problems, where it shows a significant improvement. However, a notable challenge is that the introduction of INR necessitates bi-level optimization, which potentially introduces instability and difficulty into the training process.

**Strengths:**

1.The proposed method is highly innovative.
2.The quality of writing is excellent, and the paper is easy to read.
3. As an empirical paper, the experimental results surpass the baselines.

**Weaknesses:**

1.The paper lacks research on related work. Besides Geo-FNO and FFNO, transformer-based methods can also naturally handle irregular geometric areas. The authors should at least mention these works. Here are some references for consideration [1,2,3].
2. Operator learning tasks are not computationally demanding, so the paper should explore the method's stability and report error bars (at least for some cases).
3. More explanation or supplementation is needed concerning the stability of the bi-level optimization, the difficulty of training, and why second-order meta-learning is adopted.

Overall, this is a good paper. If the authors can clarify the issues mentioned, I am inclined to accept it.

Reference
1. Transformer for Partial Differential Equations' Operator Learning (https://arxiv.org/abs/2205.13671)
2. GNOT: A General Neural Operator Transformer for Operator Learning (https://arxiv.org/abs/2302.14376)

**Questions:**

None

---

> ### Author Rebuttal · Authors · 2023-08-09
>
> We appreciate the reviewer's thoughtful feedback and have addressed the raised concerns below.
>
> ## Transformer-based methods
>
> > [...] Besides Geo-FNO and FFNO, transformer-based methods can also naturally handle irregular geometric areas. The authors should at least mention these works.
>
> We thank the reviewer for the references, we will add a discussion on these
> recent contributions and how they position w.r.t. other methods.
>
> ## Stability
>
> > Operator learning tasks are not computationally demanding, so the paper
> should explore the method’s stability and report error bars (at least for some
> cases).
>
> We have provided error bars in Table 1 of the PDF rebuttal
> page for the *Navier-Stokes* dataset, for CORAL and for the baselines. We
> compute the mean and standard deviation of the test MSE out of three runs with
> different seeds. Each seed induces a different parameter initialization for all the
> models and a different pair of train and test grids $X_{tr}$ , $X_{te}$ for the subsampling
> experiments.
> Note that the variance of all models increases when the subsampling ratio
> decreases, since the independent runs sample a different training and test grids.
>
> Additionally, training times for CORAL have been provided in Table 2 of the PDF rebuttal page.
>
> ## Optimization procedure
>
> > More explanation or supplementation is needed concerning the stability
> of the bi-level optimization, the difficulty of training, and why second-order
> meta-learning is adopted.
>
> The bi-level optimization that is referred throughout the paper as the inner-loop and outer-loop, is a crucial aspect for the success of our method. We show in Figure 1 of the PDF rebuttal page a comparison between 1st order and 2nd order meta-learning for training the modulated INRs.
>
> As can be seen, the 1st order method struggles to find a good local optimum for the shared parameters of the modulated INR. In contrast, the 2nd order meta-learning is more stable and is able to converge consistently to similarly-performing local optima across different runs.
>
> The intuition to stabilize the meta-learning training is to have a large learning rate for the inner-loop update (1e-2), and a much smaller one for the outer-loop (from 1e-4 to 1e-6). Increasing the batch size also helps stabilizing the training. Overall, this increases the number of epochs required to achieve a good reconstruction of the fields but favors a smooth training.

---

### Official Review · Reviewer_5RAL · 2023-06-23

**Soundness:** 4 excellent
**Presentation:** 4 excellent
**Contribution:** 3 good
**Rating:** 7
**Confidence:** 5

**Summary:**

This paper proposes a method for solving PDEs with neural networks in continuous space (and optionally time as well). The authors leverage the success of coordinate-based neural networks (or implicit neural representations – INRs) and formulate the problem as an operator learning one, akin to the Neural Operator family, bypassing the need for discretisation. In particular, the method is based on a typical encode-process-decode strategy:

1. *Encode*: the input condition of the PDE (i.e. the initial value in an initial value problem – IVP) is projected to the input latent vector (via gradient-based optimisation), which is then transformed (via an MLP) to a latent modulation of the input INR (SIREN), such that the INR is fitted to the input condition (auto-decoding).
2. *Process*: the input latent vector is transformed (via an MLP when projecting to a single output or via a neural ODE when solving over continuous time) to the output one.
3. *Decode*: the output latent vector is transformed (via an MLP) to a latent modulation of the output INR, which encodes the output signal and can then be queried in arbitrary spatial positions. To promote training stability, the authors employ a two-step process: First the parameters of the input/output INRs are fitted via a meta-learning strategy, similar to Dupont et al., ICML’22, using a shared INR across all input conditions and learning both the parameters of the INR and the latent vectors simultaneously and efficiently. Then, the parameters of the encode-process-decode pipeline are learnt by doing optimisation in the latent space.

The method is evaluated on several PDE tasks, showing competitive results against other continuous neural solvers.


**Strengths:**

-	**Clarity and presentation quality**. The paper is well-presented, has a good flow and guides the reader smoothly throughout the important ideas. Also, it is mathematically clean and concise. Finally, it is mostly well-contextualised with the relevant literature.
-	**Simplicity and Scalability (inference)**. Although the method is less theoretically motivated compared to Neural Operators, it builds on top of existing tools in the INR literature, making it easy to implement. In addition, forecasting happens in a compressed version of the physical process (a latent vector), which improves scalability.
-	**Novelty and Significance**. The paper proposes a new framework for continuous simulations, departing from the Neural Operator family, advancing the field further, in terms of efficiency and performance. The claims are empirically evaluated on a broad spectrum of PDE tasks (from IVPs and dynamics modelling to geometric design and optimisation) showcasing competitive generalisation performance across input conditions, resolutions and time horizons (for trajectory unrolls).


**Weaknesses:**

- **Connections with Dupont et al., ICML’22**. First off, it is important to acknowledge that the method, from a technical point of view, is largely based on the “functa” concept of Dupont et al., ICML’22. The authors do cite this work, but not as prominently as one would expect. Although the aforementioned paper focuses more abstractly on function representation learning, and less on PDE solvers, the current work is a natural follow-up, so I believe that the connections between the two should be stated more clearly.
-	**Efficiency and ease of training**. The training process might be slow since it happens in two steps. Although the second step, might be relatively fast since it happens in the latent space, the first one requires fitting INRs to all possible inputs and outputs in the training set. The meta-learning strategy might potentially alleviate this problem, but the authors have not discussed this matter. Could the authors provide more details on this step? How long does it take? Does it require extensive hyperparameter tuning? How sensitive are the results to different hyperparameters?
-	**Potential risk of overfitting (w.r.t. input/output signals not coordinates)**: Experimentally, it seems from the paper that the method generalises well across different input conditions, including the particularly challenging problem of trajectory unrolls. Let me explain why I am a bit surprised by this:

  - I am puzzled about the inductive biases of the encoding process (Eq. (1)). In particular, it is unclear what are the hidden assumptions regarding the input/output signals. Think of a CNN encoder for example – in that case, we would silently assume signal stationarity and shift invariance (due to the application of the same filter across different regions), locality, etc. On the other hand, the corresponding assumptions of the auto-decoder are unclear, even in an intuitive manner. I would appreciate it if the authors could comment on this, and it might be also useful for future readers.
   - Given the above, I am wondering how well the encoding process, and more importantly the entire pipeline, generalises across different input signals. In particular, in case the encoding process overfits, then small errors in the predictions of the latent codes (especially when unrolling trajectories where errors accumulate), might result in regions of the latent space “unknown” to the decoder, and therefore to unpredictable outcomes. To clarify this, it might be useful to discuss the distributions of the input conditions used. Are they challenging enough? How important is the Z-score normalisation (Appendix B.1.2.)?
   - My concern about this also comes from the fact that only a latent-space objective is minimised, which in principle does not rule out the possibility of small errors in the latent space to translate to large errors in the signal space (e.g. assume that the INR decoder has a large Lipschitz constant).
-	**Empirical evaluation and comparison against relevant methods**: I think there is room for improvement in the experimental section (extra discussion and comparisons) in order to make the claims more convincing. In particular,
  - It would be helpful to also include discrete neural solver baselines (e.g. based on CNNs, as in Stachenfeld, ICLR’22, or mesh GNNs, as in Pfaff et al., ICLR’21).
  - In addition, it would be nice to see a more thorough discussion regarding the comparison between continuous neural solvers (MPPDE, Neural Operators etc.). It is unclear to me why and when the current method performs better. For example, intuitively I don’t understand why MPPDE is dependent/overfits on the training grid. Have the authors considered comparing against the Graph/Multipole Graph Neural Operator methods (NeurIPS’20)? Does CORAL have any stronger inductive biases compared to Neural Operators? Could the authors elaborate on the above?
  - Regarding the dynamics modelling experiments, I have a few questions: (1) Have the authors compared against autoregressive one-step predictors (e.g. as in the MPPDE paper, or in Sanchez-Gonzalez et al., ICML’20)?  Obviously, the NeuralODE formulation allows for evaluation across arbitrary timestamps, but I am wondering if there is any disadvantage in this approach. (2) Have the authors considered comparing to discrete autoregressive solvers, such as the LE-PDE paper (NeurIPS’22)? (3) In certain papers (e.g. MPPDE and Sanchez-Gonzalez et al., ICML’20) the authors use a noise trick to stabilise trajectory unrolls and mitigate error accumulation. I am curious if the authors used it here as well, and if not why?


**Questions:**

The major questions that I would like to be clarified and discussed are included in the "Weaknesses" section. Below, I enlist a few extra minor comments:

**Minor comments**:

-	Could the authors explain their inference-time findings? Why is CORAL faster than DINO and MPPDE and slower than DeepONet and FNO? Why does CORAL scale batter (almost constant time regardless of the resolution)? Why didn’t the authors include results for FNO across all resolution setups?
-  Perhaps it would have been beneficial to also test on more challenging setups such as the turbulent regime of Navier-Stokes (potentially to identify limitations and failure cases).
- A recent paper that improves upon prior work and might be useful to also include as a baseline is the “Clifford neural layers for PDE modelling”, Brandstetter et al., ICLR’23.
-	L72: “infinite-dimensional functions“: This should be rephrased - maybe mappings between functions (infinite-dimensional vectors)?
-	L76: Regarding DeepONet, if I am not mistaken it is required to have the same observation grid only for training and not for testing.
-	L265: RK4 – maybe explain that you are referring to fourth-order Runge-Kutta.


**Limitations:**

The authors have included a section discussing the limitations of their work and were upfront about them. A few things remain to be clarified (e.g. time and tuning required for training, generalisation across input conditions, and more challenging experimental PDE setups). No foreseeable negative societal impact.

---

> ### Author Rebuttal · Authors · 2023-08-09
>
> We value the reviewer's extensive and detailed feedback and have effectively addressed the raised concerns as follows.
>
> >Connections with Dupont et al., ICML’22.
>
> The work by Dupont indeed inspired the INR part of our contribution, and will be better acknowledged in the final version.
>
> > Efficiency and ease of training.
>
> Certainly, training the INR is the most time-consuming step while time integration by the solver is fast. We provide additional details on the training time in Table 2 of the pdf rebuttal page. Note that accelerating INR training is an active research topic. Besides, we found out that training is relatively robust to the choice of hyperparameters. Key parameters include SIREN frequency $\omega_0$, code size $d_z$, and SIREN depth.
>
> > Potential risk of overfitting
>
> This is a great question which deserves further investigation. For now we can only answer intuitively. INRs encode global image representations in a spectral form [1] so that the role of the code is to modulate this frequential representation. [2] perform some experiments showing that each dimension of the code induces some frequential pattern in the reconstructed image.
> In a simplifying view, we could think of the code features corresponding to coefficients in a spectral basis ([2]). Then in CORAL the solver can be thought of as operating on this coordinate space. If the flow is smooth and regular the associated trajectory could be more easily learned.
>
> How is this achieved in CORAL ? CORAL is constrained (i) to operate in small dimensional code spaces, and (ii) optimization with meta learning is constrained for each image, to adapt the codes from a shared initial value to a final value with only a few (3 in the experiments) gradient steps. This avoids overfitting and constrains the code manifold to be smooth and low dimensional. For example in a forecasting experiment, two successive images will have close representations in the code space, and the code trajectories are deemed to be regular. This strategy enforces the robustness of the trajectories to small variations. This helps training the neural ODE and could explain the good behavior in extrapolation.
>
> [1] A Structured Dictionary Perspective on Implicit Neural Representations. Yüce et al. CVPR 2022
>
> [2] From data to functa: Your data point is a function and you can treat it like one. Dupont et al. ICML 2022.
>
> ## Baselines
>
> > Inclusion of discrete neural solver baselines.
>
> We would like to clarify that we did experiments with discrete solvers. The graph based encode-process-decode architecture was proposed in Pfaff 2021 and re-used within an updated version in MP-PDE, which is one of our baselines. These methods are discrete both for the spatial and time dimensions. We found out through preliminary experiments that MP-PDE was SOTA among GNN solvers which motivated our choice. CNNs weren't compared due to grid limitations.
>
> > Thorough discussion on continuous neural solvers comparison.
>
> MP-PDE, FNO and CORAL all perform well in many settings, and have their respective advantages, and weaknesses. Our point is that CORAL is a general framework that allows dealing (i) with multiple tasks, (ii) in a mesh-free setting, (iii) in a reduced latent space and generalizes well on multiple conditions. MP-PDE's mesh dependence hinders generalization in drastic mesh changes or sparse sub-sampling. As for FNO, being based on FFT it is limited to regular grids. Graph-Multipole was unstable and excluded from the baselines.
>
> > (1) Comparison against autoregressive one-step predictors.
>
> Yes, all the experiments involving dynamics modeling are performed with autoregressive one-step predictors for all the models.
>
> > (2) Comparison to discrete autoregressive solvers.
>
> Maybe this was not clear enough. There are two dynamics involved for forecasting. One is auto-regressive, $u_t = g(u_{t-1})$ and the second one is how we proceed from $u_{t-1}$ to $u_{t}$. The solver is involved for the latter computation: for a 4-step method like RK4 this requires 4 iterations and for a simple Euler forward, this would require 1 iteration only. Said otherwise, NODE generalizes the basic AR formulation.
>
> > (3) Noise trick for trajectory unrolls.
>
> Our NeuralODE training employed exponential sampling, initially using ground truth codes as inputs. We didn't use additional tricks.
>
> ## Minor comments
>
> > Inference-time analysis.
>
> At test time, DINO's adaptation requires 100x more optimization steps. MPPDE's computation is slow due to the KNN graph creation and slower message passing. DeepONet and FNO are faster due to forward computation only. CORAL's encoding/decoding is relatively resolution-insensitive and performed in parallel across all sensors.  Process operates on a fixed dimension independent of the resolution.
>
> FNO is fast due to FFT but cannot used on irregular grids, hence its previous omission. We included additional lin. interpo. + FNO results in Table 1 with corresponding inference time in Figure 2 of the PDF rebuttal page.
>
> > Turbulent regime of NS.
>
> Yes, we agree but this deserves a specific treatment and probably alternative models. Initial tests with large Reynolds numbers have shown an order of magnitude decrease for all the models.
>
> > Clifford neural layers for PDE modeling.
>
> This paper focuses on modeling the interactions between the physical fields and the variables of the model when they are intrinsically connected. We focused solely on model comparisons without Clifford algebra. But we agree that this would be an interesting extension for several PDE families.
>
> > L76:  DeepONet's observation grid.
>
> At test time, DeepONet requires the same number of sensors in their input to the Branch Net, preferably matching the training sensors. However, DeepONet can infer fields at any query coordinate through its trunk-net, akin to an INR.
>
> > L72: Infinite-dimensional rephrasing and L265: Clarification of RK4.
>
> Adjusted accordingly; your feedback is appreciated.

---

> > ### Comment · Reviewer_5RAL · 2023-08-20
> > **Response to Authors**
> >
> > Dear Authors,
> >
> > Thank you for your reply. Most of my concerns have been addressed. I will keep my score unchanged and vote for acceptance, as per my initial review (well-presented work, simple to implement, works well in practice, a new framework for continuous simulations).
> >
> > However, I would like to point out that the underlying reasons that lead to the success of this method, as well as its limitations, are still unclear to me. For example, the authors do mention that in the turbulent regime of NS the performance deteriorates substantially, but they have not discussed what are the input condition distributions that are currently tested, so as to understand the complexity of the tasks to be solved. Moreover, since there is no theoretical evidence and the inductive biases of the INR auto-decoding are hard to grasp, I think that there is a considerable gap in our understanding. I would encourage the authors to try to provide more intuition in an updated version (e.g. including the intuitive explanation given in the “Potential risk of overfitting” paragraph of their rebuttal) and investigate this further in the future.
> >
> > Some minor concerns that were not addressed and would be good to discuss in an updated version:
> > - Looking at Table 2 it is apparent that the INR training creates a large computational burden in the overall pipeline, with the processing in the latent space being only a small fraction of the overall training time. Could you compare the overall training times with those of the baselines?
> > - It is claimed that training is relatively robust to the hyperparameters. I would suggest adding some quantitative results in an updated version and discussing how the hyperparameters are chosen (is it by looking at the training error?).
> > - The authors have not discussed the following questions:
> >    - “To clarify this, it might be useful to discuss the distributions of the input conditions used. Are they challenging enough? How important is the Z-score normalisation (Appendix B.1.2.)?”
> >    - “My concern about this also comes from the fact that only a latent-space objective is minimised, which in principle does not rule out the possibility of small errors in the latent space to translate to large errors in the signal space (e.g. assume that the INR decoder has a large Lipschitz constant)”
> > - Could the authors clarify what “exponential sampling” is?

---

> > > ### Author Response · Authors · 2023-08-21
> > > **Response to reviewer**
> > >
> > > Dear Reviewer,
> > > Thank you for your insightful feedback and your careful consideration of our work. We're delighted to learn that most of your concerns have been addressed and that you are inclined to maintain your initial score while supporting acceptance. We will aim to offer more intuition behind the success of our method in the final version and reserve the exploration for more theoretical explanations for further investigation.
> > >
> > > ## Minor Comments:
> > >
> > > *  **Training time comparison**: In the final version, we will include a comparison of training times for all the baselines used in the dynamics modeling experiment. CORAL took slightly more time to train than DINO and MPPDE, with similar orders of magnitude in the context of  *Navier-Stokes* and *Shallow-Water*. On the other hand, FNO and DeepONet benefited from faster training.
> > >
> > > * **Selecting hyperparameters**: Recognizing the relevance of this aspect for potential users, we will incorporate a quantitative section that separately studies the impact of key hyperparameters on training and test errors. Initially, hyperparameters are determined based on training error, while validation utilizes extrapolation loss on the training set.
> > >
> > > * **Challenges with input distributions**: The input distributions originate from the PDE-pushforward of a certain distribution (e.g. gaussian for NS), i.e. rather than utilizing directly sampled random realizations as inputs at time $t=0$, we leverage solutions from numerical solvers at time $t=T_1$. We describe the conditions used to generate the data in Appendix A.
> > >
> > > * **Z-score normalization**: Due to codes being obtained from only a few gradient descent steps, their standard deviation is generally quite small. Consequently, they cannot be used as such for training Neural ODE / MLP. Normalizing the code is essential to obtain a meaningful input for the processor.
> > >
> > > * **Training in latent space**: Indeed, concerns can arise with latent-space architectures. Empirically, we observed that the above-mentioned code normalization effectively mitigates this concern, as a small loss over the normalized codes led to a small forecasting error.
> > >
> > > * **Term clarification**: Apologies for any confusion; the accurate term is "scheduled sampling with exponential decay." Appendix B.1.4 outlines its behavior.
> > >
> > > We appreciate your meticulous review and the constructive suggestions you've provided. We are dedicated to incorporating these refinements in the updated version.

---

### Official Review · Reviewer_e6Go · 2023-07-06

**Soundness:** 3 good
**Presentation:** 3 good
**Contribution:** 3 good
**Rating:** 6
**Confidence:** 3

**Summary:**

This work tried to introduce an implicit neural field-based framework for solving PDEs for different applications on general geometries. The method represents the input and output function spaces as an implicit neural representation. The method consists of a two-stage pipeline: first, the authors train two auto-decoders over the input and output function domains. Then, using paired data, they learn a mapping between the spatial latent code between domains. The paper shows results in initial value problem, dynamics, and predicting PDE solution from different geometry.

I think the paper is tackling a very important problem - generalizing operator and neural field method to solving PDE with different geometry (in the boundary/shape). And I also like the approach to this problem as they leveraged neural fields techniques to achieve discretization-free effect and operator techniques to allow relatively fast influence time. However, the presentation of the paper confuses me in several places (see weakness section) and that holds me back from full acceptance in the first read.

**Strengths:**

- The method is discretization free. It doesn't require the user to discretize the domain, which can have effect on the simulation quality. This method only need to run optimization with auto decoder, which is capable of taking partial observation and predict a continuous function that satisfies these observations.
- This method has shown advantages compared to many operator methods in that it doesn't require operations such as Fourier transformation which requires regular samples in the geometry. This advantages the method to potentially work with different domain geometry, as well as different observation density.

**Weaknesses:**

1. Can this method be used for Inverse design? The experiment named "geometry design" is a bit misleading as it hints at the designing the geometry (the domain). But to the best of my reading, the geometric design experiment is set-up to predict PDE solution from different geometric, rather than designing the domain geometry. It would be great if the authors can add a note on this in the supplementary.

2. It's not clear to me how does the method actually generalize to different geometry - I might miss something here. For example, how does the method encode where the boundary of the geometry is? It occurs to me that all functions are defined in the ambient space of physics domain. This makes it unclear to me how does the effect of domain boundary on the solution gets encoded. For example, if we have a bunch of samples (x_i, u_0(x_i)) that indicates the initial condition of a IVP problem, but we set two different boundary condition -  one circle these samples with a sphere and the other with a rectangular box. The current method seems to make the same prediction as the latent code is inferred from the set of samples samples (x_i, u_0(x_i)), but I can't believe that's the case in the real world - i.e. the boundary has no effect on the solution.

3. Justification for the requirement of simulated data. The system still requires paired data for training similar to the neural operator, which is more supervision required by the PINN or neural fields methods (which only requires the PDE definition). I wonder what's the way to justify this additional data?

4. Risk of overclaiming. I think "general geometries" has become very vaguely define throughout the paper. I'm under the impression that the method is able to solve equation with different boundary condition and different geometry of the boundary. But it's not clear to me how the current method is able to handle different boundary conditions, also it's not clear to me how does the method take into account of the different geometry of the simulation domain. For example, if we have the same initialization function, but different shape, would the latent code we obtained from test-time optimization of the auto-decoder be the same or different? If in test time, we want to change the boundary condition of this problem, how do I specify this? In general, I think the proper claim of the method is that it allows the neural operator method to work with irregular samples, but every trained model might still be restricted to solving the same set of problem (maybe the same boundary geometry) but with different initialization.


**Questions:**

See weakness section.

**Limitations:**

Section 5 discussed limitation

---

> ### Author Rebuttal · Authors · 2023-08-09
>
> We're thankful for the reviewer's helpful feedback and have addressed the raised concerns below.
>
> ## Application to inverse design
>
> > Can this method be used for Inverse design?
>
> We agree, this will be clarified in the final version. As you mention, the geometric design section in the core paper amounts to predicting PDE solutions based on diverse geometric settings. However, once trained the model could be used for "true" inverse design. This is  indicated in l.358 in the paper with preliminary experiments on the design of Airfoils shapes in Appendix D and Fig. 8.
>
> ## Operator Learning setting
>
> > Justification for the requirement of simulated data.
>
> These are two different settings targeting two different problems, considered separately in the literature. Ours is a pure data-based approach (same as for FNO, DeepONet, MP-PDE, etc), the assumption is that no prior explicit knowledge is available on the physical phenomenon (hence no PDE equation, no explicit Boundary Condition (BC), etc) and only data is available for training. The model is trained on a number of contexts (initial values etc.) and is due to generalize on different but similar contexts.
> PINNs aim at replacing traditional solvers for explicitly-known PDEs and is coined a data-free approach.  Moreover, a Vanilla PINN is trained for one single condition (initial value and boundary conditions) and does not generalize to different conditions.
>
> ## Geometries and meshes
>
> > It's not clear to me how does the method actually generalize to different geometry.
>
> This is a pure data-based approach where no prior knowledge of the PDE is incorporated into the framework. Therefore, the PDE conditions (initial or boundary) are not stated analytically and never used as such. Let's consider an Initial Value Problem (IVP) as exemplified by the cylinder scenario (Figure 1). When presented with two different obstacles, our method uses the initial condition on two distinct meshes, which include disparities at the obstacle boundary. This leads to a varying sampling for the INR encoding and one obtains a different code in each situation. The processor operating on the corresponding representation will output two different values. Our approach does not presume that the boundary has no influence on the solution. Instead, it captures these distinctions through the mesh differentiation and as result in the INR codes. This entails that for each situation, a unique code is acquired due to the employment of distinct mesh configurations at the obstacle boundary. Regarding the question of whether a model trained on a certain geometry distribution can generalize to entirely different geometries (e.g., circles vs. squares), we would like to acknowledge that this aspect was not explored within our current investigation. It remains an interesting avenue for future exploration.
>
> > Risk of overclaiming.
>
> We will clarify the claim in the paper. As indicated above, the boundary conditions are provided by the mesh definition and the corresponding sampling for learning the INR. Consider for example the airfoil (NACA-Euler) experiment in section 4.3 and Fig 1, 13, (the same holds for Fig. 14, 15). The model is trained on multiple shapes (BC) and evaluated at test time on similar but different airfoil shapes.  In this sense the model is able to handle different geometries at test time. For any new example (airfoil shape) the learned code will be different thus encoding the geometry. Note that this is also what allows us to perform inverse design (Appendix D.1).

---

> > ### Comment · Reviewer_e6Go · 2023-08-12
> > **Thanks for the response!**
> >
> > Thanks authors for the response! The replies address some of my confusion.
> >
> > In general, I think the paper proposes a system that combines the advantages of neural field representation with advantages of neural operators. This can be of interests to NeurIPS audiences. I will keep my acceptance rating.

---

> > > ### Author Response · Authors · 2023-08-18
> > > **Thank you for the comments !**
> > >
> > > We appreciate your positive feedback. Please don't hesitate to reach out if you have further suggestions or questions.

---

### Official Review · Reviewer_A6MB · 2023-07-12

**Soundness:** 3 good
**Presentation:** 3 good
**Contribution:** 2 fair
**Rating:** 5
**Confidence:** 4

**Summary:**

In the paper, the author proposed to use neural field in operator learning. In the CORAL model, it first encodes the input into some codes, apply an MLP on the codes, and then decode into the output function. Since the neural field can be continuous evaluated, the CORAL model can be applied to general geometries.

**Strengths:**

The paper uses neural presentation in operator learning which is a very natural choice and worthy investigation. The proposed method show promising results on time extrapolation and geometry design problem.

**Weaknesses:**

The overall framework is quite simple. The encode-decode structure has been studied in PCA basis [1] and Fourier basis [2], which are also applicable for general geometries. Compared to these well-studied conventional basis, neural representation usually use a higher number of parameters, which in the end, make the encoding unstable.

In the CORAL framework, the latent code is obtained using gradient descent. However it's possible to get very different latent code given the same input. Would such non-uniqueness cause any issue?

[1] Bhattacharya, Kaushik, et al. "Model reduction and neural networks for parametric PDEs." The SMAI journal of computational mathematics 7 (2021): 121-157.
[2] Fanaskov, Vladimir, and Ivan Oseledets. "Spectral neural operators." arXiv preprint arXiv:2205.10573 (2022).

**Questions:**

What is the dimension/size of the latent code?

One of the major advantage of this CORAL is time extrapolation. Do the authors have some insight why neural representation does better on this task?

In the experiment, the authors compared CORAL with another neural representation based method DINO. How does CORAL compared to DINO in the model design? Does it also use encoder and decoder? It will be great to address DINO also in the related work section.

In neural representation, many recent work use context grid and voxels to enhance the representation. Is it possible to compensate CORAL with grid-based representation?

**Limitations:**

The paper did not address its limitations.

---

> ### Author Rebuttal · Authors · 2023-08-09
>
> We appreciate the reviewer's thoughtful feedback and have addressed the raised concerns below.
>
> ## CORAL framework
> > The overall framework is quite simple.
>
>  CORAL adopts a classical encode-process-decode paradigm for learning operators. However, training an auto-decoder presents challenges, necessitating robust regularization techniques. The principal strength of CORAL lies in its auto-decoding mechanism, which is stabilized through careful optimization design: (i) CORAL encodes fields within a compact, low-dimensional code space, and (ii) leverages meta-learning optimization to adapt codes within a limited number of steps (3 in the conducted experiments). This strategic approach averts overfitting and ensures the development of a smooth, low-dimensional code manifold. We performed additional experiments to illustrate the stability of the learned encoding. Results provided in Fig. 1, in the pdf rebuttal page, show that this is indeed extremely stable.
>
> > In the CORAL framework, the latent code is obtained using gradient descent. However it's possible to get very different latent code given the same input. Would such non-uniqueness cause any issue?
>
> This is like for any non convex optimization problem, where the resulting solution from the INR method could correspond to different local optima based on the initial conditions and the optimization path taken.
> In all of our conducted experiments, one makes use of a shared encoder for the fields associated with a particular PDE along with consistent initialization and optimization techniques. As a result, for a given input, a unique solution code is obtained, and similar inputs tend to yield close  codes within the latent space. We provide the inference pseudo-code in Algorithm 4 in Appendix B.1.3.
>
> > What is the dimension/size of the latent code?
>
> The dimension of the latent code is detailed in Appendix B.1.4 in Tables 4 and 5.  We use 256 for *Shallow-Water* and 128 for all the other datasets.
>
> > One of the major advantage of this CORAL is time extrapolation. Do the authors have some insight why neural representation does better on this task?
>
> Across all models, time extrapolation is realized through an auto-regressive framework. In contrast to models like DeepONet, FNO, and MP-PDE, CORAL operates within a reduced representation space. DINO also operates in a reduced space, but we hypothesize that the CORAL optimization methodology contributes to the smoothing of the modulation space, thereby facilitating the training of the NeuralODE component. Within CORAL the dynamics' representation space is deliberately constrained, leading to a concentration of codes within a compact, low-dimensional manifold. This design choice enables comprehensive exploration of potential trajectories during training of the NeuralODE, resulting in enhanced generalization capabilities beyond the training horizon.
>
> ## CORAL vs DINO
> > How does CORAL compared to DINO in the model design?
>
>  DINO also operates within an encode-process-decode framework. However, our model diverges from DINO in several aspects encompassing design, training, and objectives. In terms of the objective, while DINO is tailored for dynamic spatio-temporal forecasting, requiring inputs and outputs to share the same space, CORAL functions as an operator capable of mapping input functional spaces to distinct functional spaces. This flexibility is demonstrated experimentally in Section 4.1 Initial Value Problem and Section 4.3 Geometric Design.
> Regarding design and training, our model exhibits key variations driven by preliminary trials. DINO employs a Multiplicative Fourier Network as its INR backbone, whereas CORAL relies on a SIREN architecture. Moreover, the optimization approach in CORAL is second-order, enabling code adaptation within a few gradient descent iterations (3 steps in contrast to the roughly 100 steps in DINO).
>
> ## Possible extensions with context-grid / voxels
> > Is it possible to compensate CORAL with grid-based representation?
>
> While it's possible to enhance CORAL's training efficiency on 3D data using grid-based representations, such as those seen in [1] or [2], it remains a challenge to derive an effective grid-based representation optimized for dynamics modeling within the CORAL framework. An alternative grid or voxel-based enhancement involves segmenting the input space and fitting distinct Neural Fields for each subspace, effectively reducing parameter count while preserving finer representation nuances [3,4]. Incorporating these strategies into CORAL could involve creating a new latent representation by concatenating latent codes obtained from individual smaller Neural Fields.
>
> References:
>
> [1] Variable Bitrate Neural Fields. Takikawa et. al. SIGGRAPH 2022
>
> [2] Instant Neural Graphics Primitives with a Multiresolution Hash Encoding.  Müller et al. SIGGRAPH 2022
>
> [3] Neural Sparse Voxel Fields, Lingjie Liu et al. Neurips 2020.
>
> [4] MINER: Multiscale Implicit Neural Representations. Saragadam et al. ECCV 2022.
>
>
> ## Limitations
>
> > The paper did not address its limitations.
>
> Limitations are discussed in Section 5 of the paper. Thanks to the reviewers' feedback, we will also discuss on the potential risks of overfitting.

---

### Author Rebuttal · Authors · 2023-08-09

We thank all the reviewers for their comments and suggestions. We carefully answered all the questions raised by each reviewer. We have also added new experimental results following the suggestions of the reviewers concerning:

* Consolidated results with error bars on *Navier-Stokes* (Reviewer K5kw)
* Stability of the training of the modulated INR with a 1st order vs 2nd order meta-learning on *Navier-Stokes*. (Reviewer K5kw)
* Additional FNO + linear interpolation results with irregular grids on *Navier-Stokes*. (Reviewer 5RAL)
* Additional inference time on FNO + linear interpolation (Reviewer 5RAL)
* Additional information on the training time of CORAL (Reviewer 5RAL)

These new results are summarized in the PDF rebuttal page.

---

### Author Response · Authors · 2023-08-19
**Discussion**

Thank you once more for your helpful comments and insightful recommendations. As we approach the discussion deadline, we kindly want to remind you of our keen interest in hearing your response and discussing the points raised in the rebuttal.

We would like to emphasize that we have included additional experiments in the rebuttal to address the questions raised in the various reviews.

Thanks again,
The Authors

---

### Decision · Program_Chairs · 2023-09-21

**Decision:**

Accept (poster)

**Comment:**

All reviewers liked the paper's approach to combine the advantages of neural field representation in the context of neural operators. While it would have been great to have better comparisons with modern U-Nets rather than the basic 2015 version that continues to remain a baseline in this domain, ultimately the paper's contribution has more to do with generalized geometries which neither FNO or U-Nets are good at handling.